# IMA peptides regulate root nodulation and nitrogen homeostasis by providing iron according to internal nitrogen status

Momoyo Ito[1], Yuri Tajima [1,8], Mari Ogawa-Ohnishi[2], Hanna Nishida [3], Shohei Nosaki [1,4], Momona Noda [1], Naoyuki Sotta [5], Kensuke Kawade[6,7,9], Takehiro Kamiya [5], Toru Fujiwara[5], Yoshikatsu Matsubayashi [2] & Takuya Suzaki [1,4] ✉

Legumes control root nodule symbiosis (RNS) in response to environmental nitrogen availability. Despite the recent understanding of the molecular basis of external nitrate-mediated control of RNS, it remains mostly elusive how plants regulate physiological processes depending on internal nitrogen status. In addition, iron (Fe) acts as an essential element that enables symbiotic nitrogen fixation; however, the mechanism of Fe accumulation in nodules is poorly understood. Here, we focus on the transcriptome in response to internal nitrogen status during RNS in *Lotus japonicus* and identify that IRON MAN (IMA) peptide genes are expressed during symbiotic nitrogen fixation. We show that *LjIMA1* and *LjIMA2* expressed in the shoot and root play systemic and local roles in concentrating internal Fe to the nodule. Furthermore, IMA peptides have conserved roles in regulating nitrogen homeostasis by adjusting nitrogen-Fe balance in *L. japonicus* and *Arabidopsis thaliana*. These findings indicate that IMA-mediated Fe provision plays an essential role in regulating nitrogen-related physiological processes.

Plants use a variety of strategies to acquire nitrogen, an indispensable macronutrient for all living organisms, thereby adapting to their surroundings[1]. In *Arabidopsis thaliana*, the genes encoding C-terminally encoded peptide (CEP) are expressed in the root when plants sense a lack of external nitrogen nutrients[2]. The produced CEP secreted peptides translocate into the shoot and are recognized by two leucine-rich repeat receptor-like kinases, CEP receptor 1 (CEPR1) and CEPR2, which triggers the production of polypeptides, CEP downstream 1 (CEPD1) and CEPD2[2,3]. The shoot-derived CEPD1/2 activates nitrate transporter 2.1 (NRT2.1) both transcriptionally and post-transcriptionally in the root when nitrate is present in the

rhizosphere[3,4]. This systemic signaling pathway from root to shoot to root contributes to nitrogen acquisition for plants to cope with fluctuating nitrate environments. Meanwhile, legumes can utilize nitrogen in the atmosphere by establishing a symbiotic relationship called root nodule symbiosis (RNS) with nitrogen-fixing rhizobia[5]. While RNS is largely beneficial for plants, it involves energy-consuming processes. Thus, plants are capable of controlling RNS depending on nitrogen availability[6]. Recent studies in two model legumes, *Lotus japonicus* and *Medicago truncatula*, have improved our understanding of the molecular mechanisms that control RNS in response to external nitrate. The central part of the mechanisms is that nodule inception

[1]Faculty of Life and Environmental Sciences, University of Tsukuba, Tsukuba, Ibaraki, Japan. [2]Division of Biological Science, Graduate School of Science, Nagoya University, Nagoya, Aichi, Japan. [3]Institute of Agrobiological Sciences, National Agriculture and Food Research Organization, Tsukuba, Ibaraki, Japan. [4]Tsukuba Plant-Innovation Research Center, University of Tsukuba, Tsukuba, Ibaraki, Japan. [5]Graduate School of Agricultural and Life Sciences, The University of Tokyo, Tokyo, Japan. [6]Division of Symbiotic Systems, National Institute for Basic Biology, Okazaki, Aichi, Japan. [7]School of Life Science, The Graduate University for Advanced Studies (SOKENDAI), Okazaki, Aichi, Japan. [8]Present address: Rhelixa Inc., Tokyo, Japan. [9]Present address: Graduate School of Science and Engineering, Saitama University, Saitama-city, Saitama, Japan. ✉e-mail: suzaki.takuya.fn@u.tsukuba.ac.jp

(NIN)-like protein (NLP) transcription factors (TFs) regulate the positive and negative expression of RNS-related genes[7–9]. Lj/MtNLP1 can additionally modulate RNS by controlling nitrate uptake through *Lj/MtNRT2.1* expression[10,11]. Thus, a greater understanding of the mechanisms in response to external nitrogen nutrients is underway. In addition to such mechanisms responsive to external nitrogen nutrients, it is conceivable that plants regulate physiological processes in response to internal nitrogen status; however, the latter mechanisms remain mostly elusive.

Iron (Fe) is an essential micronutrient to sustain numerous biochemical processes. Hence, plants tightly regulate Fe transport, distribution, and homeostasis according to environmental Fe availability[12]. In *A. thaliana*, FER-like iron deficiency-induced transcription factor (FIT) is a key TF in regulating plant responses to Fe deficiency that interacts with basic helix–loop–helix subgroup Ib (bHLH Ib) TFs, including bHLH38/39/100/101[13–16]. POPEYE (PYE) belongs to the bHLH IVb subgroup and also regulates plant responses to Fe deficiency[17]. These bHLH TFs activate the expression of numerous genes required for plants to cope with Fe deficiency. The gene expression of *FIT*, *bHLH Ibs*, and *PYE* are dependent on upstream bHLH IVc TFs, including bHLH34/104/105/115[15,16]. When plants fulfill their Fe needs, a putative Fe sensor BRUTUS (BTS) acts as an E3 ubiquitin ligase to promote the degradation of bHLH IVc proteins, thereby inhibiting plant responses to Fe deficiency[18,19]. Iron man (IMA) peptides, nonsecreted small proteins, are synthesized both in the shoot and the root under Fe-deficient conditions. The resulting IMA peptides interact with BTS and interfere with its function[20,21], which allows FIT, bHLH Ibs, and PYE to activate the expression of the Fe-deficiency-responsive genes. Reciprocal grafting experiments indicate that IMA peptides in the shoot have a systemic function in the Fe deficiency response, whereas those in the root have a local function[20].

In RNS, Fe has an essential role in symbiotic nitrogen fixation to function as a cofactor of leghemoglobin, an oxygen-carrying protein required for symbiotic nitrogen fixation, and nitrogenase that catalyzes nitrogen fixation[5,22,23]. Legumes have a mechanism to accumulate Fe in infected cells of nodules using Fe transporters, including natural resistance-associated macrophage protein 1 and proteins similar to *A. thaliana* vacuolar iron transporters (VITs) that specifically function in nodules[24–26]. The loss-of-function of these Fe transporters attenuates symbiotic nitrogen fixation, indicating Fe plays an essential function in RNS. In *M. truncatula*, the nodule-specific cysteine-rich (NCR) 247 peptide confiscates haem groups to promote Fe uptake by rhizobia colonized in nodules[27]. Although there are few reported cases of TFs involved in Fe signaling in legumes, soybean GmbHLH57 and GmbHLH300, which are orthologous to AtFIT and AtbHLH38/39/100/101, are shown to be involved in the regulation of Fe uptake[28]. In particular, GmbHLH300 regulates Fe uptake in nodules by controlling the expression of *yellow stripe-like 7*, encoding a putative Fe transporter[29]. Despite these progresses in our understanding of the mechanisms by which plants cope with Fe deficiency and those by which plants uptake Fe into nodules, our knowledge is incomplete about the molecular mechanisms of how Fe signaling is activated by rhizobial infection and how Fe is provided to nodules from external and internal sources during RNS.

Here, we show that *LjIMA* genes are expressed depending on internal nitrogen status during RNS. Functional analyses of LjIMA1 and LjIMA2 peptides reveal a mechanism by which Fe is systemically and root-locally accumulated in nodules. We further show that IMA peptides regulate nitrogen homeostasis by adjusting nitrogen-Fe balance. These findings indicate that IMA-mediated Fe provision plays an essential role in the regulation of nitrogen-related physiological processes.

## Results

### *LjIMA* genes are upregulated during symbiotic nitrogen fixation
To obtain gene expression profile in response to internal nitrogen status during RNS in *L. japonicus*, we conducted transcriptome analysis

in non-inoculated, wild-type (WT) and ΔNifH rhizobia-inoculated conditions. In the case of inoculation with ΔNifH rhizobia with defective nitrogen-fixing activity, the plants could not benefit from symbiotic nitrogen fixation, regardless of nodule development (Supplementary Fig. 1). In the growth conditions, no external nitrogen nutrients were supplied to focus on internal nitrogen status. At 11 days after inoculation (dai), the plants inoculated with ΔNifH rhizobia were defective in nitrogen-fixing activities, but they showed no obvious differences in plant growth from non- or WT rhizobia-inoculated conditions (Supplementary Fig. 1). In contrast, the causal relationship between defective symbiotic nitrogen fixation and plant growth was evident at 18 dai (Supplementary Fig. 1a). Given that gene expression occurred prior to morphological and metabolic changes, whole shoot tissues at 11 dai were collected for transcriptome analysis. As a result, we identified 1965 genes whose expression was rhizobia-inducible and nitrogen fixation-inducible (Supplementary Fig. 2 and Supplementary Data 1). We then hypothesized that shoot-derived signaling molecules might transmit internal signals between organs and screened for genes that encode putative small peptides consisting of less than 100 amino acid residues. We consequently identified a set of peptide genes, which are homologous to *A. thaliana IMA* genes[20,30], showed a common gene expression pattern; a database search found eight *LjIMA* genes five of which, namely *LjIMA1*, *LjIMA2*, *LjIMA4*, *LjIMA5*, and *LjIMA8*, were expressed in the shoot during symbiotic nitrogen fixation (Fig. 1a and Supplementary Fig. 3a).

### LjIMA1/2 are required for the establishment of RNS
Among the *LjIMA* genes, we focused on *LjIMA1*, which had the highest gene expression, and its paralogue, *LjIMA2*, for further functional analysis (Fig. 1a and Supplementary Fig. 3b). Knockout plants of the genes were generated by the CRISPR–Cas9 system and their phenotypes in RNS were analyzed (Supplementary Fig. 4). Every single mutant showed no obvious phenotype, whereas the *Ljima1/2* double mutants exhibited an increase in nodule number and a decrease in nodule size (Fig. 1b–d). In addition, nitrogen-fixing activity in *Ljima1/2* was lower than in WT, suggesting that nodule function was attenuated in *Ljima1/2* (Fig. 1e). Thus, LjIMA1 and LjIMA2 are likely to have redundant functions for the positive regulation of nodule function. At an earlier stage of RNS, such as 5 dai, the number of infection threads, an indicator of rhizobial infection foci[31], was increased 1.5-fold in *Ljima1/2* compared to WT (Fig. 1f), suggesting LjIMA1/2 have an additional role in RNS like the control of rhizobial infection process.

As *LjIMA1/2* expressions were initially detected in the shoot, we then performed grafting experiments to assess whether LjIMA1/2 has systemic roles in the control of RNS. The nodule and shoot phenotypes of *Ljima1/2* were restored when WT was used as a scion, suggesting that LjIMA1/2 in the shoot systemically affects RNS (Fig. 2a–d). Unexpectedly, normal nodule formation and shoot growth occurred when *Ljima1/2* and WT were used for scion and rootstock, respectively. Hence, it is possible that in addition to their systemic roles of LjIMA1/2 in the shoot, LjIMA1/2 in the root locally functions to regulate RNS.

The result of grafting experiments suggested the possibility that LjIMA1/2, both in the shoot and root, might function in RNS, whereas their gene expression was initially observed in the shoot (Fig. 1a). Therefore, we next determined the temporal expression patterns of *LjIMA1/2* in the shoot and root during RNS. High expression levels of both genes were detected in the shoot at later stages of RNS, including 11 dai (Supplementary Fig. 5a, b). Furthermore, *LjIMA1* was found to be expressed at relatively earlier stages of RNS, in which nitrogen fixation was seemingly yet to be active. In particular, the *LjIMA1* expression at 5 dai was observed even when plants were inoculated with ΔNifH rhizobia (Supplementary Fig. 5a); which suggests that the *LjIMA1* expression at such an earlier stage of RNS is due to rhizobial infection rather than nitrogen fixation. In the root, the *LjIMA1/2* expression levels tended to increase with time after rhizobial inoculation

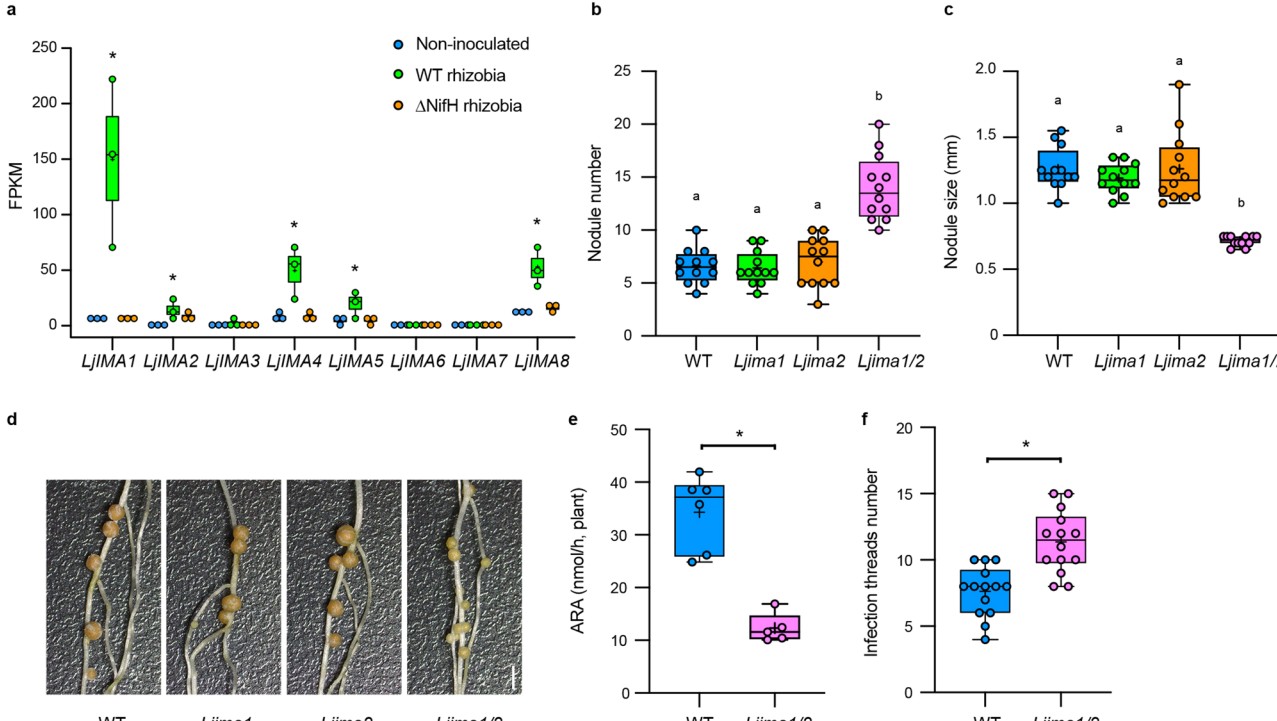

**Fig. 1 | LjIMA1/2 regulates root nodule symbiosis. a** Expression levels of *LjIMA* genes in the shoot in non-inoculated, wild-type (WT) and ΔNifH rhizobia-inoculated 11 d after inoculation (dai) without external nitrogen nutrients. Whole shoots of *L. japonicus* WT MG-20 plants were used for transcriptome analysis (*n* = 3 independent pools of shoots derived from three plants). Fragments per kilobase of exon per million reads mapped (FPKM) of each gene are shown. **b**–**d** Nodulation phenotypes of each plant 21 dai. **b** Total nodule number (*n* = 12 plants). **c** Maximum nodule diameter of nodules on the root of each plant (*n* = 12 plants). **e** Acetylene reduction activity (ARA) of WT (*n* = 6 plants) and *Ljima1/2* (*n* = 5 plants) plants 21 dai. **f** The number of infection threads (ITs) formed in WT and *Ljima1/2* plants 5 dai (*n* = 14

plants). ITs were identified by GFP signals derived from the transgenic rhizobia constitutively expressing *GFP*. Scale bar, 2 mm. Centerlines in the boxplots show the medians, and box limits indicate the 25th and 75th percentile. The whiskers go down to the smallest value and up to the largest. Scatterplots show individual biological replicates as dots. In **a**, asterisks indicate a statistically significant difference compared with non-inoculated conditions (*P* < 0.05, one-way analysis of variance (ANOVA) followed by multiple comparisons). In **b**, **c**, different letters indicate statistically significant differences (*P* < 0.0001, one-way ANOVA followed by multiple comparisons). In **e**, **f**, asterisks indicate a statistically significant difference (*P* < 0.01, by a two-sided Welch's *t* test).

(Supplementary Fig. 5c, d). The expression of *LjIMA1/2* in the root at 11 dai, as well as in the shoot, was likely dependent on nitrogen fixation because ΔNifH rhizobia did not contribute to the gene expression at this stage. In addition, analysis of *LjIMA1_pro:GUS* plants showed that *LjIMA1* was expressed in leaf vascular tissues and nodules (Supplementary Fig. 5e, f).

### LjNIN regulates *LjIMA1* expression

In exploring the upstream mechanisms of *LjIMA1* expression, we noticed there is a semi-palindromic nucleotide sequence 0.6 kb upstream of the initiation codon on the *LjIMA1*, which resembles the NIN-binding sequence (NBS) and nitrate-responsive *cis*-element (NRE)[9,32]. NIN, a master regulator of nodule formation, regulates its target gene expression by binding to NBS, which is structurally similar to NRE bound by NLPs[9,32,33]. An electrophoresis mobility shift assay (EMSA) showed that LjNIN, LjNLP1, and LjNLP4 bound to the NBS/NRE-like sequence on the *LjIMA1* promoter (Fig. 3a), although the two LjNLPs had less binding capacity to the nucleotide sequence than LjNIN. Furthermore, a transactivation assay using mesophyll proto-plasts of *L. japonicus* suggested that the LjNIN and LjNLP4 regulate gene expression by binding to the NBS/NRE-like sequence on the *LjIMA1* promoter (Fig. 3b). To further examine the LjNIN-mediated *LjIMA1* expression, we created transgenic hairy roots in which *LjNIN* was overexpressed in the absence of rhizobia. We found that the *LjNIN* expression resulted in the expression of *LjIMA1*, not *LjIMA2* (Fig. 3c, d). In addition, in *Ljnin* mutants, rhizobia-dependent *LjIMA1/2* expression was suppressed both in the shoot and root at 3 dai (Fig. 3e). Given that

LjNIN functions predominantly in the root[33], the reduced *LjIMA1* expression in the root is likely a direct effect of the *Ljnin* mutation, whereas that in the shoot may be due to the fact that the signals necessary for the gene expression are not transmitted from root to shoot.

### LjIMA1/2 has an essential role in Fe accumulation in nodules to establish RNS

To further elucidate the function of LjIMA1/2, we analyzed the plants with hairy roots that overexpressed *LjIMA1/2*. Gene ontology analysis using transcriptome data of these plants showed the involvement of LjIMA1/2 in Fe-related processes (Supplementary Data 2). Compared to control roots, several putative Fe-related genes were upregulated by *LjIMA1/2* overexpression. These included genes encoding Fe transporters (iron-regulated transporter 1, LjIRT1 and LjVIT), a Fe translocator (nicotianamine synthase 2, LjNAS2), a Fe reductase (ferric reductase oxidase 2, LjFRO2), and a TF for Fe signaling (LjFIT1) (Fig. 4a). In addition, the *LjIMA1/2* overexpression in the root caused a distinctive leaf phenotype under normal growth conditions containing 10 μM Fe (Supplementary Fig. 6a). The leaf phenotypes had a similarity to those that became obvious when WT plants were fed with a high concentration of Fe (5 mM) (Supplementary Fig. 6a, b), which implied that *LjIMA1/2* overexpression caused an excessive response to Fe. In our growth conditions, 1 mM Fe was not a toxic concentration for plants, as we did not observe the leaf symptom that was visible in 5 mM Fe conditions (Supplementary Fig. 6b). In addition, the Fe amount in the root was increased by *LjIMA1/2* overexpression in normal growth

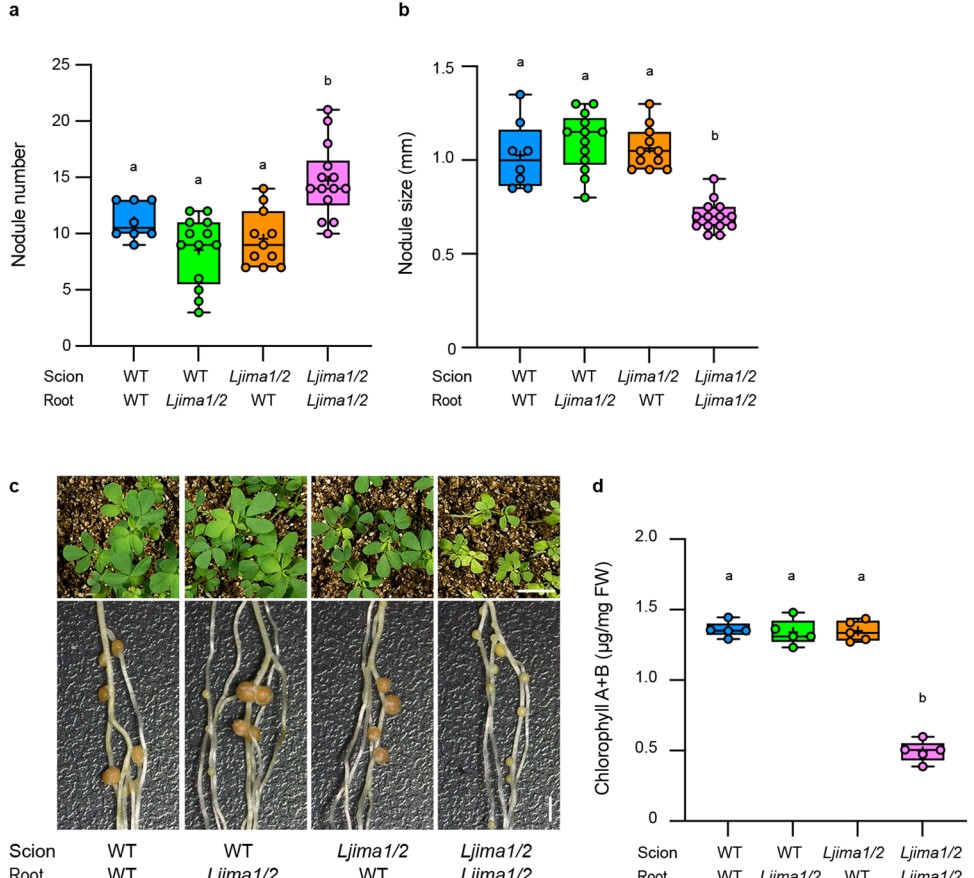

**Fig. 2 | LjIMA1/2 in the shoot and root regulate root nodule symbiosis.**
**a**–**c** Nodulation and shoot phenotypes of grafted plants (scion/root) 21 dai. **a** Total nodule number of WT/WT ($n = 8$ plants), WT/*Ljima1/2* ($n = 13$ plants), *Ljima1/2*/WT ($n = 11$ plants) and *Ljima1/2*/*Ljima1/2* ($n = 14$ plants). **b** Maximum diameter of nodules on the root of of WT/WT ($n = 8$ plants), WT/*Ljima1/2* ($n = 13$ plants), *Ljima1/2* / WT ($n = 11$ plants) and *Ljima1/2* / *Ljima1/2* ($n = 14$ plants). **d** Shoot chlorophyll

contents of grafted plants 21 dai ($n = 5$ plants). Scale bars, 1 cm (upper in **c**), 2 mm (lower in **c**). Centerlines in the boxplots show the medians, and box limits indicate the 25th and 75th percentile. The whiskers go down to the smallest value and up to the largest. Scatterplots show individual biological replicates as dots. In **a**, **b**, **d**, different letters indicate statistically significant differences ($P < 0.05$, one-way ANOVA followed by multiple comparisons).

conditions (Supplementary Fig. 6c). These results suggest that LjIMA1/2 have a positive role in Fe-related processes, including Fe uptake, translocation, and signaling. *LjIMA1/2* overexpression attenuated nodule formation (Supplementary Fig. 6d). The supply of high concentrations of Fe (5 mM) attenuated mature nodule formation without affecting total nodule number (Supplementary Fig. 6e, f). It is thus probable that excessive Fe has a negative effect on RNS.

To verify the defect of RNS in *Ljima1/2* in terms of Fe involvement, we quantified the Fe amount. Although the amount of Fe in the shoot was almost unaffected, that of in the root increased by rhizobial inoculation in WT but not in *Ljima1/2* (Fig. 4b). The observations indicate that LjIMA1/2 are required for Fe accumulation in the root during RNS. Of note, staining of Fe in nodules showed that *Ljima1/2* was unable to accumulate Fe in nodules (Fig. 4c). We further examined the effect of Fe supply on the *Ljima1/2* RNS phenotype. The nodulation phenotype of *Ljima1/2* was restored by Fe supply in a dosage-dependent manner (Fig. 4d–f). Especially in the presence of 1 mM Fe, nodule formation in *Ljima1/2* was indistinguishable from that of WT. In addition, we found that rhizobial inoculation induced *LjNAS2* expression in an LjIMA1/2-dependent manner in the root (Fig. 4g). Consistent with this, accumulation of nicotianamine in nodules was dependent on LjIMA1/2 (Fig. 4h). The *LjIRT1* expression level at 5 dai was higher in *Ljima1/2* than WT, whereas the expression of *LjIRT1* and *LjFRO2a* were unaffected by rhizobial inoculation, implying that *Ljima1/2* has a higher demand for Fe than WT (Supplementary Fig. 7). Although *LjIMA1/2* expressions were activated by rhizobial infection, they were

downregulated by Fe supply (Fig. 4i). A more detailed analysis of the relationship between Fe concentrations and *LjIMA1/2* expression levels in RNS showed that 0.1 mM Fe caused a decrease in *LjIMA1/2* expression in the shoot (Supplementary Fig. 8). In contrast, a significant decrease in *LjIMA1* expression in the root was observed at 0.5 mM Fe. Given that *LjIMA1/2* expression may reflect Fe demand, the Fe demand may partly differ between the shoot and the root in RNS. Collectively, our data indicate that LjIMA1/2 acts according to plant Fe demand and has an essential role in Fe accumulation in nodules to establish RNS.

Since *Ljima1/2* has increased nodule number, LjIMA1/2 may be related to the autoregulation of nodulation (AON), a mechanism that negatively regulates nodule number[34,35]. Overexpression of *LjIMA1/2* reduced the number of nodules in *hypernodulation aberrant root formation 1* (*har1*), an AON-defective mutant, although the effect was not as pronounced as when they were overexpressed in WT (Supplementary Figs. 6d and 9a). Furthermore, a reduction in the number of mature nodules was observed by the supply of 1 mM Fe in *har1* (Supplementary Fig. 9b, c). These results support the possibility of the implication of LjIMA1/2 and Fe in AON.

## LjIMA1/2 are required for nitrogen homeostasis
We have shown that *LjIMA1/2* expression was induced during symbiotic nitrogen fixation. The fact that nitrogen fixation produces ammonium led us to postulate the possibility that LjIMA1/2 might have roles in response to external nitrogen nutrients. To test the possibility, we analyzed *LjIMA1/2* expressions by treatment of nitrogen nutrients in

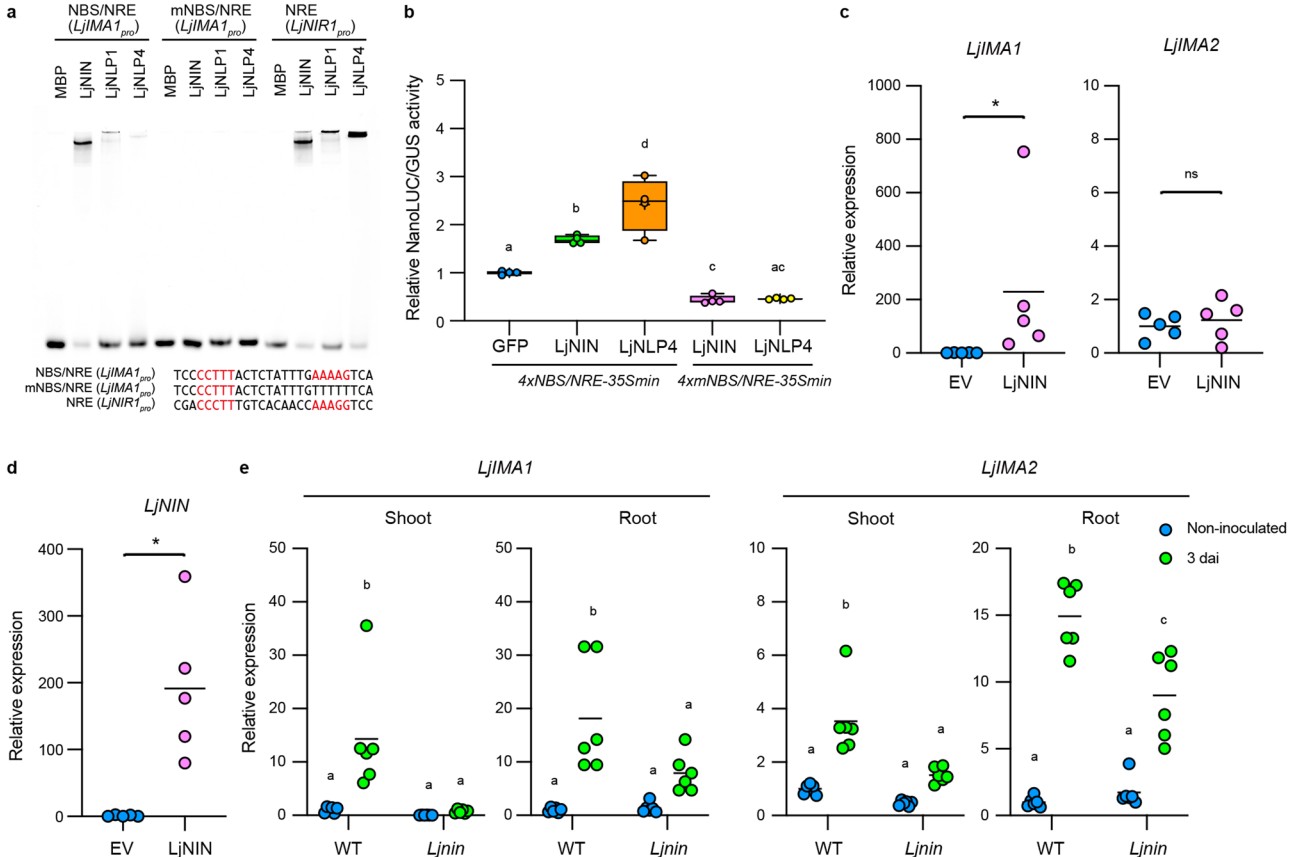

**Fig. 3 | LjNIN regulates *LjIMA1* expression. a** Electrophoresis mobility shift assay (EMSA) showing LjNIN, LjNLP1, or LjNLP4 binding to the NIN-binding sequence (NBS)/nitrate-responsive *cis*-element (NRE), mutated NBS/NRE (mNBS/NRE) or NRE on the *LjIMA1* or *LjNIR1* promoter. Nucleotide sequences forming semi-palindromic structures are shown in red. MBP-LjNIN (551–878), MBP-LjNLP1 (573–903), or MBP-LjNLP4 (564–976) recombinant proteins, consisting of an RWP-RK and a PB1 domain, were incubated with the FAM-labeled DNA probe (Supplementary Data 3). NRE on the *LjNIR1* promoter was used as a positive control. Three independent experiments were conducted to obtain similar results. **b** Transactivation of *4xNBS/NRE-35Smin:NanoLUC* and *4xmNBS/NRE-35Smin:NanoLUC* in *L. japonicus* mesophyll protoplasts (*n* = 4 independent pools of protoplasts). NanoLUC activity was measured relative to *35Spro:GUS* activity. Transactivation data were normalized to the condition in which *GFP* was expressed. **c, d** RT-qPCR analysis of *LjIMA1*, *LjIMA2*, and *LjNIN* expression in the transgenic hairy roots carrying the construct that constitutively expressed *LjNIN* in the absence of rhizobia. Transgenic roots were

identified by GFP signals and collected (*n* = 5 independent pools of hairy roots), and total RNA was isolated for cDNA synthesis. RT-qPCR data were normalized to the data of roots where an empty vector (EV) was introduced. **e** RT-qPCR analysis of *LjIMA1/2* expression in the WT and *Ljnin* mutants. Non-inoculated or 3 dai shoots or roots were collected (*n* = 6 independent pools of shoots or roots derived from three plants). RT-qPCR data were normalized to WT non-inoculated conditions. The expression of *LjUBQ* was used as the reference. Centerlines in the boxplots show the medians, and box limits indicate the 25th and 75th percentile. The whiskers go down to the smallest value and up to the largest. Scatterplots show individual biological replicates as dots. In **c-e.**, bars indicate mean values. In **b, e**, different letters indicate statistically significant differences (*P* < 0.05, one- or two-way ANOVA followed by multiple comparisons). In **c, d**, asterisks indicate a statistically significant difference (*P* < 0.01, by a two-sided Mann–Whitney test). ns means not significant.

the absence of rhizobia. The expression of *LjIMA1/2* was induced by nitrogen nutrients in the shoot, especially including ammonium (Fig. 5a, b). NLP-dependent nitrate-responsive gene expression is shown to be a quick response to nitrate, and LjNLP4 predominantly functions in the root[8,36]. We thus examined *LjIMA1/2* expression in the root 6 h after nitrate treatment and found that they were quickly induced in the root by nitrate in an LjNLP4-dependent manner (Supplementary Fig. 10). The observation is consistent with the previous results that LjNLP4 bound to the NBS/NRE-like sequence on the *LjIMA1* promoter (Fig. 3a, b). Furthermore, insufficient expression of *LjIMA1/2* due to deficient internal nitrogen was restored by external nitrogen (Supplementary Fig. 11).

To further clarify the LjIMA1/2 functions in nitrogen response, we next examined nitrogen-dependent plant growth. Plants were grown under normal growth conditions with altering nitrate concentration in the absence of rhizobia. Unlike WT, the positive effect of nitrate on plant growth was less significant in *Ljima1/2* (Supplementary Fig. 12a, b). In addition, *Ljima1/2* accumulated higher levels of nitrate than WT in a

dosage-dependent manner (Supplementary Fig. 12c). Regarding the ammonium contents, it remained at low levels with the supply of nitrate and ammonium, and there was no difference between WT and *Ljima1/2* (Supplementary Fig. 12d). These observations suggest that *Ljima1/2* mutation attenuates nitrate-dependent plant growth and nitrate assimilation. Nitrate promoted plant growth of WT and inhibited that of *Ljima1/2* in a dosage-dependent manner (Supplementary Fig. 12a, b), which might be related to the fact that *LjIMA1* had nitrate dosage-dependent expression (Supplementary Fig. 12e). The nitrate-affected incomplete plant growth, increase in nitrate contents, and decrease in chlorophyll contents in *Ljima1/2* were restored by Fe supply (Fig. 5c–f). Furthermore, Fe downregulated the nitrogen-dependent *LjIMA1/2* expression (Supplementary Fig. 13a). Similarly, Fe downregulated the *LjIMA1* promoter activity (Supplementary Fig. 13b). These results indicate that LjIMA1/2 are involved in the adjustment of nitrogen–Fe balance that is necessary for nitrogen homeostasis.

We next investigated the influence of nitrogen–Fe balance in RNS and the involvement of LjIMA1/2. In the absence of Fe, consistent with

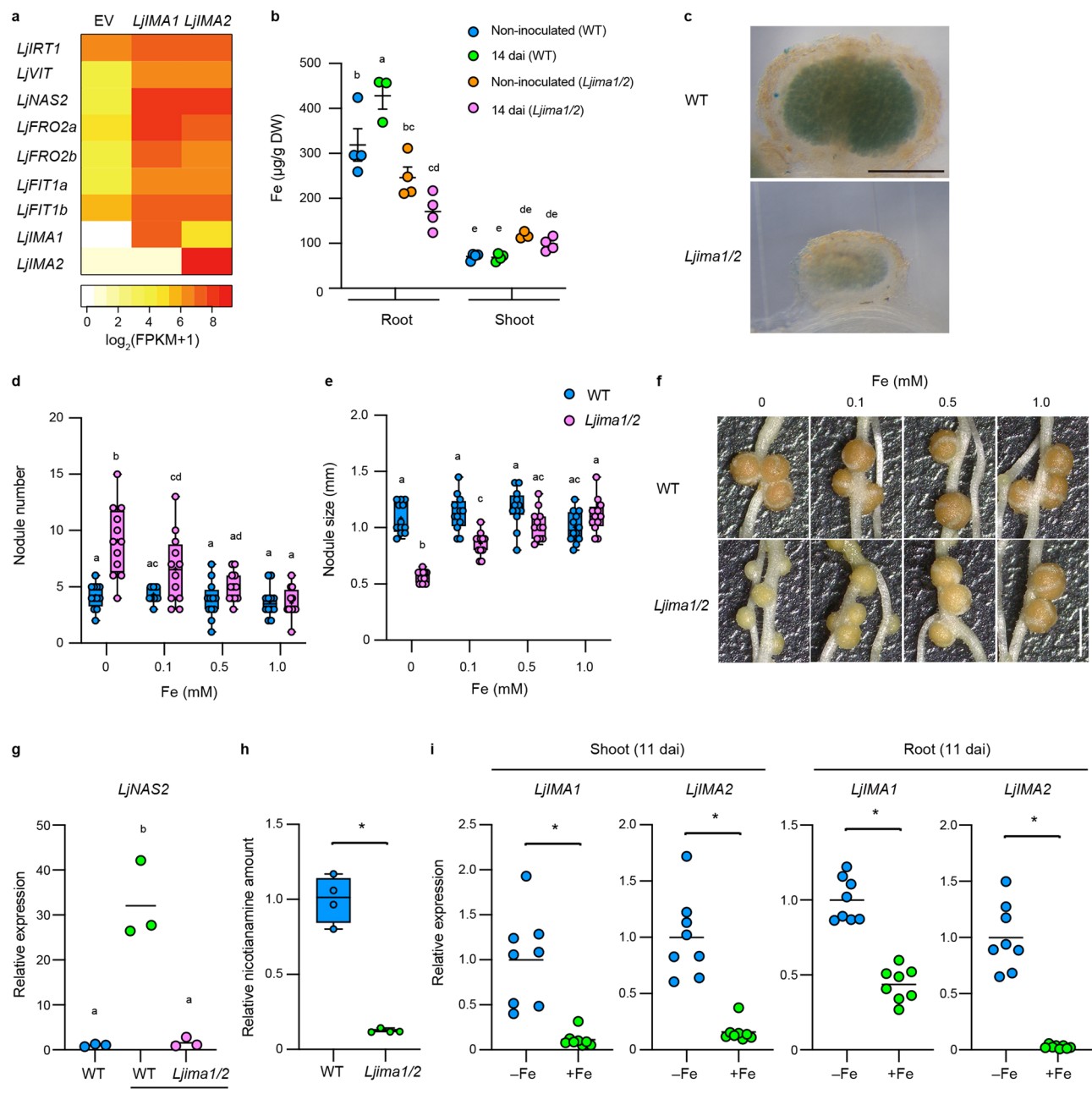

**Fig. 4 | LjIMA1/2-mediated Fe provision is required for root nodule symbiosis.**
**a** Transcriptome analysis of Fe-related genes in response to *LjIMA1/2* over-expression. Transcriptome data were obtained from the WT plants with transgenic hairy roots carrying control empty vector (EV), *LjUBQ_pro:LjIMA1* or *LjUBQ_pro:LjIMA2* constructs 5 dai (*n* = 3 independent pools of hairy roots from three plants). The expression patten of *LjIRT1* (Lj4g3v1428550), *LjVIT* (Lj4g3v0336390), *LjNAS2* (Lj3g3v1048990), *LjFRO2a* (Lj0g3v0105009), *LjFRO2b* (Lj0g3v0360999), *LjFIT1a* (Lj0g3v0107279), *LjFIT1b* (Lj3g3v1113460), *LjIMA1* (Lj2g3v3071890), and *LjIMA2* (Lj0g3v0238729) are shown. **b** Fe amounts of non-inoculated or 14 dai shoots or roots in WT and *Ljima1/2* (*n* = 3 or 4 independent pools of shoots or roots derived from 12 plants). **c**. Perls Prussian Blue Fe staining of 14 dai nodules in WT and *Ljima1/2*. Three independent experiments were conducted to obtain similar results.
**d**, **f** Nodulation phenotypes of WT and *Ljima1/2* 14 dai in different Fe concentrations. **d** Total nodule number (*n* = 12 plants). **e** Maximum nodule diameter of nodules on the root of each plant (*n* = 12 plants). **g** RT-qPCR analysis of *LjNAS2* expression in WT and *Ljima1/2*. Non-inoculated (0 dai) or 5 dai roots were collected (*n* = 3 independent pools of roots derived from three plants). RT-qPCR data were

normalized to WT 0 dai conditions. **h** Relative nicotianamine amount in WT and *Ljima1/2* nodules 21 dai (*n* = 4 independent pools of roots derived from 12 plants). Data were normalized to nicotianamine amount in WT. **i** RT-qPCR analysis of *LjIMA1/2* expression in WT. Shoots or roots of 11 dai plants with 1 mM Fe (+Fe) or without Fe (−Fe) were collected (*n* = 8 independent pools of shoots or roots derived from three plants). RT-qPCR data were normalized to −Fe conditions. The expression of *LjUBQ* was used as the reference. Fe (III)-EDTA was used for the Fe source. Scale bars, 500 μm in (**c**) and 2 mm in (**f**). Centerlines in the boxplots show the medians, and box limits indicate the 25th and 75th percentile. The whiskers go down to the smallest value and up to the largest. Scatterplots show individual biological replicates as dots. In **b**, error bars indicate SEM. In **g**, **i**, bars indicate mean values. In **b**, **d**, **e**, different letters indicate statistically significant differences ($P < 0.05$, two-way ANOVA followed by multiple comparisons). In **g**, different letters indicate statistically significant differences ($P < 0.01$, one-way ANOVA followed by multiple comparisons). In **h**, **i**, asterisks indicate a statistically significant difference ($P < 0.01$, by a two-sided Welch's *t* test or Mann–Whitney test).

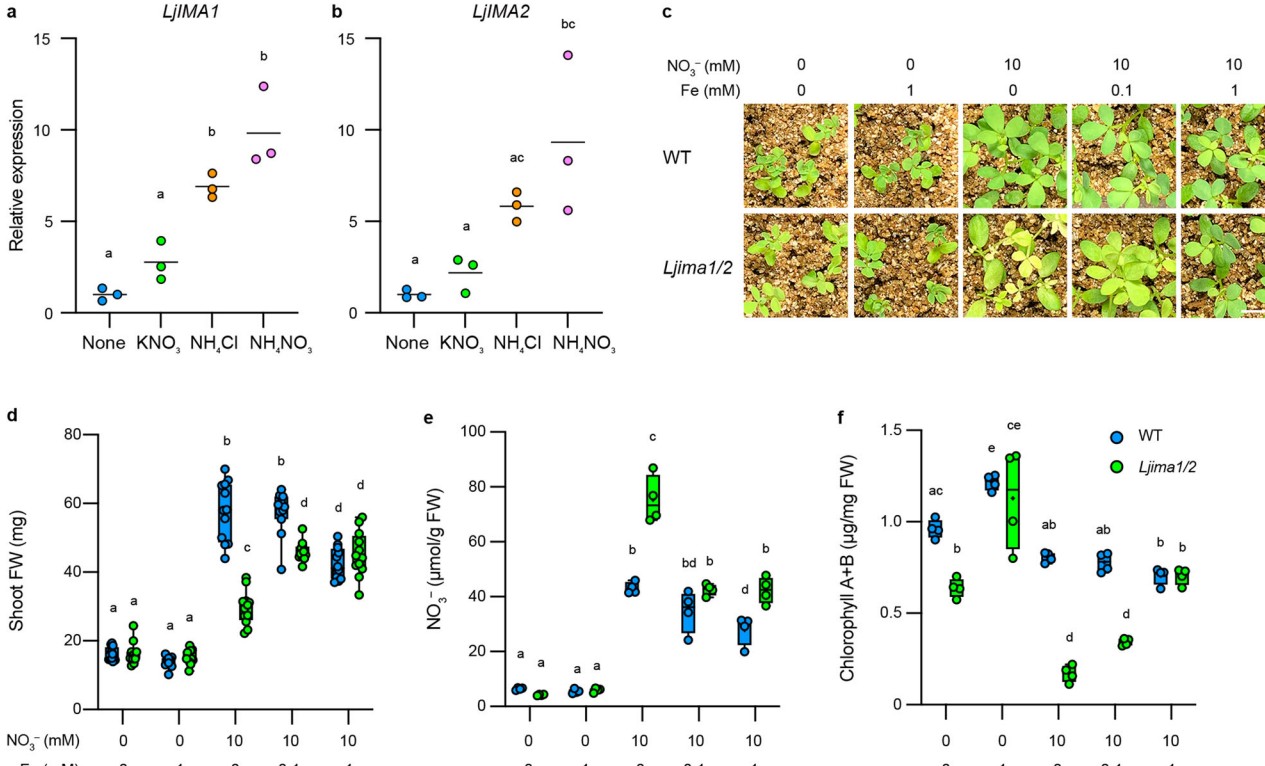

**Fig. 5 | LjIMA1/2 is required for nitrogen homeostasis depending on the nitrogen–Fe balance. a**, **b** RT-qPCR analysis of *LjIMA1/2* expression in response to nitrogen nutrients. Shoots were collected from WT plants treated with 5 mM $KNO_3$, 5 mM $NH_4Cl$, or 2.5 mM $NH_4NO_3$ for 2 d in the absence of rhizobia ($n = 3$ independent pools of shoot from three plants). RT-qPCR data were normalized to non-treated conditions. The expression of *LjUBQ* was used as the reference. **c**–**f** Shoot phenotypes, shoot FW ($n = 11$ or 12 plants) and shoot nitrate ($n = 4$ independent pools of shoot derived from 2 or 3 plants) and shoot chlorophyll contents ($n = 4$ independent pools of shoot derived from 2 or 3 plants) of WT and *Ljima1/2* grown in different nitrogen–Fe ratios for 17 d after germination in the absence of rhizobia. $KNO_3$ and Fe (III)-EDTA were used for nitrogen and Fe sources, respectively. Scale bar, 1 cm. Centerlines in the boxplots show the medians, and box limits indicate the 25th and 75th percentile. The whiskers go down to the smallest value and up to the largest. Scatterplots show individual biological replicates as dots. In **a**, **b**, bars indicate mean values. In **a**, **b**, different letters indicate statistically significant differences ($P < 0.05$, one-way ANOVA followed by multiple comparisons). In **d**–**f**, different letters indicate statistically significant differences ($P < 0.05$, two-way ANOVA followed by multiple comparisons).

previous reports, a high concentration of nitrate (10 mM) inhibited nodule formation in WT (Supplementary Fig. 14a–c)[8–10]. A low concentration of nitrate (1 mM) did not affect nodule formation in WT, but it attenuated nodule formation in *Ljima1/2* (Supplementary Fig. 14a–c), suggesting that *Ljima1/2* is hypersensitive to nitrate in RNS. In the presence of 1 mM nitrate, the nitrate levels in *Ljima1/2* accumulated to the same level as when WT was fed with 10 mM nitrate (Supplementary Fig. 14d, e). Thus, in *Ljima1/2*, excessive nitrate accumulation may be the cause of the inhibition of nodule formation. The hypersensitive response to nitrate and the excessive nitrate accumulation was restored by Fe supply (Supplementary Fig. 14). Therefore, it is likely that nitrate inhibition of nodulation is controlled by LjIMA1/2-mediated adjustment of nitrogen–Fe balance.

### IMA-mediated control of nitrogen homeostasis is conserved in Arabidopsis

Finally, to validate the general function of the IMA peptides revealed in *L. japonicus*, we tested whether similar peptides had a role in nitrogen response in *A. thaliana*. *AtIMA* gene expression was investigated under normal growth conditions with 50 µM Fe, varying only the nitrogen nutrient concentration. We consequently found that the expression of *AtIMA1/2/3* genes was induced in the shoot and root by a high concentration of nitrogen nutrient (50 mM) (Supplementary Fig. 15a–f). In the high nitrogen condition, *AtIRT1* and *AtFRO2* expression were also induced (Supplementary Fig. 15g,h), suggesting that high nitrogen nutrients triggered a Fe-deficiency response. Furthermore, consistent with the results of *L. japonicus*, the nitrogen-induced *AtIMA3*

expression was suppressed in the presence of relatively high Fe concentrations (Supplementary Figs. 13a and 15i). Thus, *Lj/AtIMA* gene expression is likely regulated by the ratio of nitrogen to Fe.

We then investigated the function of some *AtIMA* genes in nitrogen response. To this end, *Atima3* single and *Atima1/2/3* triple mutants were generated by the CRISPR–Cas9 system (Supplementary Fig. 4). WT and *Atima3* showed comparable plant growth in 5 and 50 mM nitrate conditions (Fig. 6a). In contrast, plant growth was reduced in the case of 50 mM nitrate treatment compared to 5 mM in *Atima1/2/3* mutants (Fig. 6a). In *Atima3* and *Atima1/2/3* mutants, 50 mM nitrate reduced chlorophyll contents compared to 5 mM (Fig. 6b). As reported in the past[20], *Atima3* exhibited growth defects with reduced chlorophyll contents in Fe-deprived conditions (Fig. 6c, d). Moreover, *Atima3* accumulated more nitrate than WT in Fe-deprived and nitrate-rich conditions (Fig. 6e). Therefore, Lj/AtIMA peptides have essentially the same function with respect to nitrogen response. Overall, it is reasonable to conclude that IMA peptides generally regulate nitrogen homeostasis by adjusting the nitrogen–Fe balance.

### Discussion

In this study, we discovered that *LjIMA1/2*, expressed in response to internal nitrogen status, regulates RNS by providing Fe to the nodules. To date, IMA peptides have been characterized exclusively for their function in Fe-deficiency response in *A. thaliana*[20,21,30,37]. In RNS, Fe acts as an essential element that enables symbiotic nitrogen fixation;[5,22,23] however, how Fe accumulates in nodules is poorly understood. We showed that LjIMA1/2 functions in response to plant Fe demand and

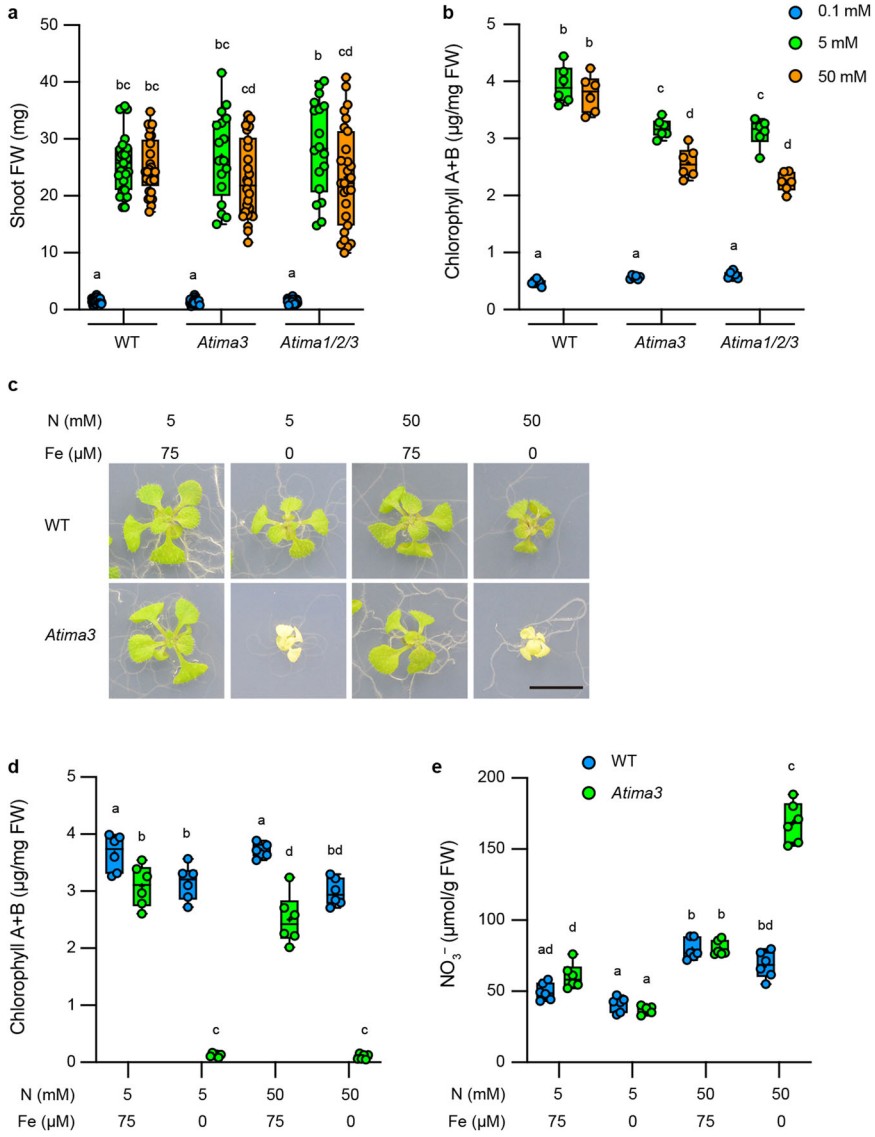

**Fig. 6 | AtIMA3 is required for nitrogen homeostasis depending on nitrogen–Fe balance. a, b** Shoot FW (*n* = 17–40 plants) and chlorophyll contents (*n* = 6 independent pools of shoot derived from three plants) of WT, *Atima3*, and *Atima1/2/3* plants. Plants were grown on agar plates with 1/2 MS medium containing different concentrations of nitrate for 14 d. **c–e** Effects of different nitrogen–Fe ratios on plant growth, nitrate contents (*n* = 6 independent pools of shoot derived from three plants), and chlorophyll contents (*n* = 6 independent pools of shoot derived from three plants). Plants were grown on agar plates with 1/2 MS medium containing different concentrations of nitrate and Fe for 14 d. Scale bar, 1 cm. Centerlines in the boxplots show the medians, and box limits indicate the 25th and 75th percentile. The whiskers go down to the smallest value and up to the largest. Scatterplots show individual biological replicates as dots. In **a, b, d, e**, different letters indicate statistically significant differences (*P* < 0.05, two-way ANOVA followed by multiple comparisons).

plays an essential role in the accumulation of Fe into nodules. The expression of several key genes in Fe signaling, including *LjIRT1* and *LjFRO2*, whose homologous genes in *A. thaliana* are involved in external Fe uptake and utilization[12], are induced by constitutive expression of *LjIMA1/2*. As far as we examined, *LjIRT1* and *LjFRO2* are not strongly activated in RNS, nor is their expression dependent on *LjIMA1/2*. In contrast, *NAS2*, encoding a nicotianamine synthase, shows rhizobia-inducible and *LjIMA1/2*-dependent expression. Indeed, LjIMA1/2 is required for the accumulation of nicotianamine in nodules. In *M. truncatula*, *MtNAS2* is shown to be required for symbiotic nitrogen fixation[38]. Given the primary role of nicotianamine as an Fe translocator[12], there may be a mechanism that *LjIMA1/2* expressed in the shoot and root play systemic and local roles in concentrating internal Fe to the nodule. This mechanism could be a reasonable plant adaptive strategy to establish RNS in Fe-limited environments. High expression levels of *LjIMA1/2* in the later stages of RNS, such as 11 dai

are likely due to the fact that more Fe is required for nitrogen fixation. Moreover, *LjIMA1* expression is induced in relatively early stages of RNS, when nitrogen fixation is yet to be active. This may reflect a mechanism in which plants prepare for prompt Fe accumulation at the sites of incipient nodule primordia. It is not fully understood how Fe in the nodule is further transported to and utilized by rhizobia. In some legumes, such as *M. truncatula*, that form indeterminate nodules, NCR247 peptide regulates Fe uptake by rhizobia through binding to haem groups[27]. Since NCR peptides are absent in some legumes that form determinate nodules[39], including *L. japonicus*, it is unclear whether the finding of NCR247 can be extended to every legume species. The next challenge is to determine the detailed role of LjIMA peptides in the nodule, which includes investigating whether they have the additional function of Fe transport to rhizobia. Although we showed the possibility that LjIMA1/2 and Fe may be involved in AON, the details of this relationship remain unclear. The possibility of crosstalk

between Fe and AON is particularly intriguing, and further clarification of the genetic relationship between LjIMA1/2 and AON factors will provide new insights into the role of AON.

Under our growth conditions, *L. japonicus* WT plants do not show any obvious symptoms of Fe deficiency in the absence of external Fe (0 mM Fe conditions), nor symptoms of Fe excess even at a concentration of 1 mM Fe. The apparent normality of the plant growth in the absence of Fe may be due to the use of internal Fe stored in the seed and/or Fe originally contained in vermiculite. Anyway, *L. japonicus* likely has a wider range of Fe concentrations that it can adapt to than *A. thaliana*. Elucidating the molecular mechanism behind this can be an interesting research topic because it may lead to a further understanding of the diverse plant's strategy for adapting to the Fe environment.

In our understanding of the evolutionary basis of RNS, it is thought that RNS emerged by partly co-opting some pre-existing regulatory systems. For example, a key signaling cascade used in arbuscular mycorrhizal symbiosis was recruited to the early nodulation signaling[40,41]. In addition, some factors involved in lateral root formation were diverted to nodule organogenesis[42,43]. Nevertheless, other aspects of the evolutionary basis of RNS, including nodule function, remain elusive. Our data propose a scenario in which IMA peptides, originally involved in Fe uptake and signaling, might have been co-opted to RNS to provide an essential nutrient with nodules for enabling nodule function. In the case of *LjIMA1*, the presence of an NLP-binding NRE on its promoter might have been a predisposition that facilitated the direct regulation by NIN. The fact that LjNIN regulates *LjIMA1* expression indicates a direct molecular link between the Fe acquisition mechanism and an important nodulation signaling pathway. Given the LjNIN-dependent *LjIMA1* expression in early RNS stages such as 3 dai, when nitrogen fixation is not active, NIN may be involved in accumulating Fe at the sites of incipient nodule primordia. To determine whether NIN is additionally involved in the regulation of nitrogen–Fe balance, its role needs to be clarified at later RNS stages, especially in nodules with nitrogen-fixing activity.

Nitrogen and Fe are, respectively, essential nutrients for plant growth, but the molecular mechanisms of their interrelationships remain enigmatic. It has been recently shown that OsNLP4 regulates the nitrogen–Fe balance by controlling the expression of Fe signaling genes, thereby promoting tillering in rice[44]. This observation is likely to support our finding that LjNLP4 directly regulates the expression of *LjIMA1*. We propose that, as a common and novel function of IMA peptides in plants, they regulate nitrogen homeostasis by adjusting the nitrogen–Fe balance. *IMA* genes are expressed when nitrogen levels are high relative to Fe. The resulting IMA peptides could activate Fe-related processes, which contributes to adjusting the nitrogen-Fe balance. In this regard, *IMA* gene expression appears to behave like a sensor of nitrogen-Fe balance. It is currently unknown why nitrate is highly accumulated in *ima* mutants of *L. japonicus* and *A. thaliana* when nitrate levels are high relative to Fe. Since Fe is required for the action of nitrate reductase like nitrogenase, the conditions may cause a relative lack of Fe and malfunction of this enzyme, resulting in retardation of the nitrate assimilation process. Meanwhile, it is an open question how the *LjIMA1/2* expression is induced in the shoot by rhizobial infection or nitrogen nutrients. If Fe demand signals mediate between these stimuli and gene expression, it is probable that a TF similar to AtbHLH105 could be involved in regulating *LjIMA1/2* expression in these contexts, as is the case in *A. thaliana*[21]. Future elucidation of the regulatory mechanisms of *IMA* expression in various tissues and phenomena involving nitrogen will improve the understanding of the molecular link between nitrogen and Fe signaling.

## Methods

### Plant materials and growth conditions

The Miyakojima MG-20 ecotype of *L. japonicus* was used as the WT plant[45]. A description of *Ljnin*, *Ljnlp4/nrsym1*, and *har1* plants was published previously[8,46]. Plants were grown with or without *Mesorhizobium loti* MAFF 303099 in autoclaved vermiculite or on 1% agar plates with Broughton and Dilworth (B&D) solution[47] under a 16 h light/8 h dark cycle at 24 °C. ΔNifH strain of *M. loti* (clone ID: 14T03d07), which was previously created by the signature-tagged mutagenesis[48], was provided from Legume Base (Miyazaki University). For the observation of infection threads, plants were inoculated with *M. loti* MAFF 303099 constitutively expressing *GFP*[49]. For grafting experiments, seeds were germinated on 1% agar plates in a growth cabinet (24 °C dark for first 2 d, 24 °C 16 h light/8 h dark cycle for next 2 d). Then, seedlings were cut perpendicularly at the hypocotyls with a scalpel blade. A shoot scion was then sliced at an angle and inserted into a short vertical slit made on a rootstock. Grafted plants were sandwiched by two filter papers saturated with sterilized water in plastic plates for 3 d, and transferred to vermiculite pots containing rhizobia.

The Columbia (Col-0) ecotype of *A. thaliana* was used as the WT plant. Plants were grown on 1.3% agar plates with respective nutrients that have basically the same concentrations as those of 1/2 Murashige–Skoog (MS) medium under a 16 h light/8 h dark cycle at 22.5 °C. The concentration of nitrogen nutrients was controlled by changing the amount of $KNO_3$.

### Stable and hairy root transformation

To create *Ljima1*, *Ljima2*, and *Ljima1/2* knockout plants by CRISPR–Cas9 system, gRNAs were designed using the CRISPR-P 2.0 program[50] (Supplementary Fig. 3). Two gRNAs were used to target a gene, and they were cloned to pMR203_AB and pMR203_BC, gRNA cloning vectors, and then to pMR285_AD, a binary vector[51] by a standard ligation method using a DNA Ligation Kit (Takara) and several restriction enzymes. To create $LjIMA1_{pro}$: *GUS* plants, $PcUBQ_{pro}$: *Cas9* cassette in pMR285 was removed, and $GUS\text{-}RBCS_{ter}$ cassette[8] was introduced. Then, a *LjIMA1* promoter fragment, including a 2.9 kb upstream region of the initiation codon, was amplified by PCR and inserted upstream of *GUS*. For stable transformation of *L. japonicus*, seeds were germinated on germination medium (1/2x Gamborg's B5 salt mixture (Wako), 1/2× Gamborg's vitamin solution (Sigma), 1% sucrose, 1% agar) in a growth cabinet (24 °C dark for first 2 d, 24 °C 16 h light/8 h dark cycle for next 2 d). *A. tumefaciens* GV3101 MP90RK strains harboring each construct were streaked on YEP plate with appropriate antibiotics for 2 d at 28 °C. Seedlings were placed in the *A. tumefaciens* suspension, and then their hypocotyls were cut into about 3 mm pieces. The hypocotyl pieces were placed onto the top of pilled filter papers saturated with co-cultivation medium (1/10× Gamborg's B5 salt mixture, 1/10× Gamborg's vitamin solution, 0.2 μg ml⁻¹ BAP, 0.05 μg ml⁻¹ NAA, 5 mM MES (pH 5.2), 20 μg ml⁻¹ acetosyringone, pH 5.5) and were incubated in a growth cabinet (21 °C dark) for 6 d. After that, the hypocotyl pieces were transferred to callus induction medium (1× Gamborg's B5 salt mixture, 1× Gamborg's vitamin solution, 2% sucrose, 0.2 μg ml⁻¹ BAP, 0.05 μg ml⁻¹ NAA, 10 mM $(NH_4)_2SO_4$, 0.3% phytagel, 12.5 μg ml⁻¹ meropen, 15 μg ml⁻¹ Hygromycin B, pH 5.5) and were incubated in a growth cabinet (24 °C 16 h light/8 h dark cycle) for 2–3 w. The hypocotyl pieces were transferred to a fresh callus induction medium every 1 w. When calluses became more than 1 mm in size, they were detached from the hypocotyls and transferred onto callus medium without hygromycin B, and were incubated for 3–7 w in a growth cabinet (24 °C 16 h light/8 h dark cycle) until leaf primordia became visible. The calluses were transferred onto a new medium every 1 w. The calluses with leaf primordia then were transferred to shoot elongation medium (1× Gamborg's B5 salt mixture, 1× Gamborg's vitamin solution, 2% sucrose, 0.2 μg ml⁻¹ BAP, 0.3% phytagel, 12.5 μg ml⁻¹ meropen, pH 5.5), and incubated until their shoot length became about 1 cm. Individual shoots were detached from calluses and transferred to root induction medium (1/2× Gamborg's B5 salt mixture, 1/2× Gamborg's vitamin solution, 1% sucrose, 0.5 μg ml⁻¹ NAA, 0.4%

phytagel, 12.5 µg ml⁻¹ meropen, pH 5.5), and incubated for 8 d. Then, they were transferred to a root induction medium without NAA and cultivated until their root length became about 2–3 cm. The resultant transgenic plants were transplanted into soils for further cultivation. Transgenic plants with homozygous mutations were used for analysis (Supplementary Fig. 3).

For *LjIMA1*, *LjIMA2*, or *LjNIN* overexpression in *L. japonicus* hairy roots, the *LjUBQ~pro~-NOS~ter~* cassette was amplified by PCR from an original vector[9] and was cloned to pCAMBIA1300-GFP by the In-Fusion (Clontech) reaction. Then, the coding sequence (CDS) of *LjIMA1*, *LjIMA2*, or *LjNIN* was amplified by PCR and was cloned downstream of *LjUBQ~pro~* by the In-Fusion reaction. For hairy root transformation, seeds were germinated on the germination medium described above in a growth cabinet (24 °C dark for first 2 d, 24 °C 16 h light/8 h dark cycle for next 2 d). *A. rhizogenes* AR1193 strains harboring each construct were streaked on YEP plate with appropriate antibiotics for 2 d at 28 °C. Seedlings were placed in the *A. rhizogenes* suspension and then cut at the base of the hypocotyls. The seedlings with cotyledons were transferred onto hairy root medium (1× Gamborg's B5 salt mixture, 1× Gamborg's vitamin solution, 2% sucrose, 1% agar) and were grown in a growth cabinet (24 °C dark for first 1 d, 24 °C 16 h light/8 h dark cycle for next 2 d). Then, the plants were transferred onto fresh hairy root medium containing 12.5 µg ml⁻¹ meropen and were grown for 7 d in a growth cabinet (24 °C 16 h light/8 h dark cycle)[52]. Transgenic roots were identified by GFP fluorescence, and the plants with transgenic hairy roots were used for further experiments.

To create *Atima1/2/3* knockout plants by CRISPR-Cas9 system, gRNAs were designed using the CRISPR-P 2.0 program[50] (Supplementary Fig. 3). A gRNA was used to target a gene, and *U6-26~pro~* and gRNAs were cloned into pKIR1.0[53,54]. Transgenic Arabidopsis plants were generated by the standard Agrobacterium-mediated transformation using the floral dip method. Transgenic plants with homozygous mutations were used for analysis (Supplementary Fig. 3).

## Acetylene reduction assay

The nitrogenase activity of nodules was indirectly determined by measuring the acetylene reductase activity[8]. Nodulated roots detached from intact plants were put into 20 mL vials. Subsequently, acetylene was injected into the vials. After incubation for 10 or 20 min, the amount of ethylene produced was measured using a GC-2014 (Shimadzu).

## Electrophoresis mobility shift assay

Recombinant proteins were previously prepared[10]. For preparing the probes, DNA fragments (Supplementary Data 3) were labeled with carboxyfluorescein (FAM). The labeled DNA fragments were purified on the Superdex 200 increase 10/300 GL column (Cytiva). The purified DNA fragments (0.25 µM) and poly (dI–dC) (50 ng/µL) were mixed with the purified proteins in a buffer (10 mM Tris−HCl pH 7.5, 100 mM KCl, 100 mM NaCl, 1 mM DTT, 2.5% glycerol and 5 mM MgCl₂), and incubated at 25 °C for 30 min. The mixtures were loaded on a 10% polyacrylamide gel, and fluorescence was detected using LuminoGraph III WSE-6300 (ATTO).

## Transactivation assay

*NanoLUC* and *RBCS~ter~* were respectively amplified by PCR from pNL1.1.PGK (Promega) and an original vector[8], and were simultaneously cloned to pUC19 by the In-Fusion reaction to make a *NanoLUC-RBCS~ter~* cassette. A fragment was artificially synthesized, in which four tandem repeats of the 40-bp region of *LjIMA1~pro~* harboring NBS/NRE or mNBS/NRE sequence were fused with 35 S minimal promoter (35Smin). The *4xNBS-35Smin* or *4xmNBS-35Smin* fragments were cloned upstream of *NanoLUC*. Other constructs, *LjUBQ~pro~*: *GFP* and *LjUBQ~pro~*: *LjNIN* were previously created[9]. Mesophyll protoplasts of *L. japonicus* were isolated from WT plants grown for 16 d with 10 mM KNO₃. The

seedlings, including leaves, stems and cotyledons, were chopped by a razor blade and were incubated in an enzyme solution (20 mM MES (pH 5.7), 1.5% Cellulase R10 (Yakult), 2% Macerozyme R10 (Yakult), 0.4 M mannitol, 20 mM KCl, 10 mM CaCl₂, and 0.1% BSA) at 24 °C in the dark for 5 h. For transformation, 100 µl protoplasts solution in MMG solution (4 mM MES (pH 5.7), 0.4 M mannitol, and 15 mM MgCl₂) and 15 µl DNA mixture solution were mixed with 115 µl of PEG solution (40% PEG4000, 0.2 M mannitol, and 100 mM CaCl₂). Then, protoplasts were incubated overnight at 24 °C in the dark in WI solution (4 mM MES (pH 5.7), 0.5 M mannitol, and 20 mM KCl) and harvested by centrifugation. Harvested protoplasts were resuspended in extraction buffer (100 mM phosphate buffer (pH 7.2), 5 mM DTT, and Complete Protease Inhibitor Cocktail (Roche), and were frozen in liquid nitrogen and warmed at 37 °C for 5 min. For GUS assays, 100 µL cell lysate was mixed with 100 µL GUS assay buffer (100 mM phosphate buffer (pH 7.2), 1 mM EDTA, 0.1% Triton X-100, 5% glycerol, 1 mM 4-methylumbelliferyl-β-D-glucuronide (4-MUG), and 17% methanol). The mixture was incubated for 1 h at 37 °C, and then 800 µL 0.2 M Na₂CO₃ was added. Fluorescence was measured using a Synergy LX (Biotek). The Luminescence of NanoLUC was detected using the Nano-Glo Luciferase Assay System (Promega) with the Synergy LX. For the transformation of protoplasts, an equal amount of DNA (2.5 µg each) of effector, reporter, and internal control plasmids was used. *35S~pro~*: *GUS* was used as the internal control plasmid.

## RNA-seq analysis

Total RNA was isolated using the PureLink Plant RNA Reagent (Invitrogen). Libraries were prepared using a NEBNext Ultra II RNA Library Prep Kit from Illumina (New England Biolabs) following the manufacturer's instructions and sequenced using a Novaseq 6000 (Illumina) instrument with the 150-bp paired-end sequencing protocol. RNA-seq reads were mapped to the *L. japonicus* MG-20 genome version 3.0 using HISAT2 (ver. 2.1.0) with the default parameters[55]. Mapped reads were then assembled using StringTie (ver.1.3.5)[56]. Up- or down-regulated genes were identified with edgeR (ver. 3.40.2), and fragments per kilobase exon per million reads (FPKM) values were computed after trimmed-mean-of-M-values normalization[57]. Gene ontology (GO) enrichment analysis was performed using the GO term Molecular Function with the LotusBase[58] GO Enrichment (https://lotus.au.dk/go/enrichment).

## Gene expression analysis

The primers used for PCR are shown in Supplementary Data 3. Total RNA was isolated from respective tissues using the PureLink Plant RNA Reagent (Invitrogen) or the Plant Total RNA Mini Kit (Favorgen Biotech). First-strand cDNA was prepared using the ReverTra Ace qPCR RT Master Mix with gDNA Remover (Toyobo). RT-qPCR was performed using a 7900HT Real-Time PCR system (Applied Biosystems) or a CFX Opus 384 Real-Time PCR system (BIO-RAD) with a THUNDERBIRD SYBR qPCR Mix (Toyobo) following the manufacturer's instructions.

## Measurement of Fe contents

Plant tissues were dried in an oven at 70 °C for at least 24 h. Approximately 10–20 mg of the dried samples were measured and then put in heat-resistant Teflon tubes. 1 mL of HNO₃ (density 1.38, for boron determination; Wako) was added to each sample, and the tubes were heated with heat blocks overnight at 100 °C. After the HNO₃ in the tubes had completely evaporated, 300 µL of HNO₃ was added and heated at 100 °C until the samples solidified. Finally, 300 µL of hydrogen peroxide (H₂O₂) was added to the samples and heated at 100 °C to obtain white pellets. The resultant pellets were dissolved in 1 mL of 0.08 N HNO₃. The solution was diluted 10 times with 0.08 N HNO₃ and subjected to inductively coupled plasma mass spectrometry (ICP-MS) (Agilent 7800 ICP-MS). Indium (In) was added as an internal control, and measurements were normalized by the signals of In.

## Perls Prussian blue Fe staining

Root segments with nodules were placed in Perls solution (4% HCl, 4% potassium ferrocyanide) deaerated for 15 min and incubated at normal pressure for 30 min. Samples were rinsed several times with sterilized water. For observation of cross-sections of nodules, the stained samples were fixed with 4% paraformaldehyde and 0.25% glutaraldehyde in 50 mM phosphate buffer for 16 h and dehydrated in a graded ethanol series. Subsequently, they were embedded in Technovit 7100 (Kulzer).

## Measurement of nicotianamine contents

Plant tissues were frozen in liquid nitrogen immediately after sampling and stored at −80 °C prior to liquid chromatography–tandem mass spectrometry (LC–MS/MS) analysis for nicotianamine. The samples were crushed and suspended in 500 μL 80% methanol containing 0.1% formic acid and 1 pmol/μL $^{13}C^{15}N$ L-Valine as an internal standard. The supernatant was collected after centrifugation, evaporated in a centrifugal concentrator, and re-suspended in 20 μL 0.1% formic acid and 25 mM EDTA[59]. In total, 2 μL aliquot was used for LC-MS/MS analysis based on the Triple TOF 5600 system (AB Sciex).

## Measurement of nitrate and ammonium contents

Plants grown in respective conditions were washed with sterilized water and frozen in liquid nitrogen. Each sample was crushed using a TissueLyser II (Qiagen), and 10 μl of sterilized water at 80 °C was added to per 1 mg of sample weight, and vortexing was performed every 5 min for 20 min at 100 °C. The sample was then chilled on ice and spun down. The supernatant was stored as an extraction solution at −80 °C. In total, 20 μl of 0.05% salicylic acid in sulfuric acid or sulfuric acid was added to 5 μl of extraction solution in a tube, which was vortexed, spun down, and left at room temperature for 20 min. Then, 500 μl of 8% NaOH in sterilized water was added, and the mixture was vortexed until it became clear. Nitrate content in the solution was determined by absorbance at 410 nm using the Synergy LX. For ammonium contents, samples were prepared for nitrate measurements and determined using a LabAssay Ammonia (Fujifilm)[60] with the Synergy LX.

## Measurement of chlorophyll contents

Seedlings grown under each experimental condition were washed with sterilized water and frozen with liquid nitrogen. Each sample was crushed using a TissueLyser II, and 1.6 mL of chilled acetone was added. Samples were centrifuged at 20,000$g$ at 4 °C for 5 min. Then, 800 μL supernatant was mixed with 200 μL sterilized water on ice. Absorbances at 646.6 nm, 663.6 nm, and 750 nm in solution were measured using a DU 800 Spectrophotometer (Beckman). The following formula determined the amount of chlorophyll A and B in the solution[61].

$$\text{chlorophyll A (μg/ml)} = 12.25 \times (OD_{663.6} - OD_{750}) - 2.85 \times (OD_{646.6} - OD_{750})$$

$$\text{chlorophyll B (μg/ml)} = 20.31 \times (OD_{646.6} - OD_{750}) - 4.91 \times (OD_{663.6} - OD_{750})$$

## Statistical analysis

Statistical analysis was performed using GraphPad Prism version 9 (GraphPad Software). Normality was checked using the Shapiro–Wilk test and $P > 0.05$ was considered as normal distribution. The $F$-test was used to test whether the variances of the two populations were equal. Appropriate methods were chosen according to the nature of the data. The criterion of $P < 0.05$ means a statistically significant difference in this study.

## Reporting summary

Further information on research design is available in the Nature Portfolio Reporting Summary linked to this article.

## Data availability

Gene IDs described in this study are described in Supplementary Data 4. Sequence data from this article can be found in the GenBank/EMBL data libraries under the following accession number: LjIMA1 (LC770146), LjIMA2 (LC770147), LjIMA3 (LC770148), LjIMA4 (LC770149), LjIMA5 (LC770150), LjIMA6 (LC770151), LjIMA7 (LC770152), LjIMA8 (LC770153). Data from the short reads from RNA-seq analysis were deposited in the DNA Data Bank of Japan Sequence Read Archive under the accession numbers DRA016472 [https://ddbj.nig.ac.jp/resource/bioproject/PRJDB16002] and DRA016483 [https://ddbj.nig.ac.jp/resource/bioproject/PRJDB16001]. Source data are provided in this paper.

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

## Acknowledgements

We thank Legume base (National BioResource Project, Miyazaki University) for providing ΔNifH strain of *M. loti*; Tomoko Mori (National Institute for Basic Biology, NIBB) and Trans-Omics Facility, NIBB Trans-Scale Biology Center for technical support of LC–MS/MS analysis; the Advanced Analysis Center of National Agriculture and Food Research Organization (NARO) for use of analysis servers; Masayoshi Kawaguchi (NIBB), Shigetaka Yasuda (Nara Institute of Science and Technology) and Kei Hiruma (The University of Tokyo) for technical supports. This research was supported by Ministry of Education, Culture, Sports, Science and Technology KAKENHI grants (JP20H05908 and JP23H02495 to T.S.); JST Mirai Program (JPMJMI20E4) to T.S; NIBB Collaborative Research Program (23NIBB456) to K.K; the Cooperative Research Grant of the Plant Transgenic Design Initiative by Gene Research Center, Tsukuba-Plant Innovation Research Center, University of Tsukuba (#2303) to H.N.

## Author contributions

M.I. and T.S. conceived the project and designed the experiments. M.I., M.N. and T.S. performed the experiments on *L. japonicus*. Y.T., M.O.-O. and Y.M. performed the experiments on Arabidopsis. M.I. and H.N. analyzed transcriptome data. S.N. performed EMSA. N.S., T.K. and T.F. measured Fe amounts. K.K. measured nicotianamine amounts. M.I. and T.S. interpreted the results and wrote the paper.

## Competing interests

The authors declare no competing interests.
