## [Peer Review File · Nature Communications]

IMA peptides regulate root nodulation and nitrogen homeostasis by providing iron upon internal nitrogen statusReviewer #1 (Remarks to the Author):

Iron is an essential micronutrient to sustain numerous biochemical processes, including symbiotic nitrogen fixation. Hence, plants tightly regulate its transport, distribution, and homeostasis according to environmental iron availability. The transcription factor PYE is a crucial genetic component in regulating plant responses to iron deficiency. PYE activates the expression of numerous genes required to cope with this nutritional status. Once plants fulfill their iron needs, the E3 ubiquitin ligase BRUTUS, an iron sensor, interacts with PYE, promotes its degradation, and inhibits plant responses to iron deficiency.

IRON MAN (IMA) peptides are synthesized under iron-deficient conditions and interact with BRUTUS, which allows PYE to activate the expression of the iron-deficiency-responsive genes. Hence, IMA peptides play a crucial role in iron uptake. Despite this progress in our understanding of how plants cope with iron deficiency, we still need to understand how root nodules obtain the appropriate concentration of iron to sustain the symbiotic nitrogen fixation process. In this manuscript, the authors performed different experiments to answer this question experimentally in the model legume *Lotus japonicus*. The authors found that the expression of LjIMA1 and LjIMA2 is increased in shoots and roots from plants inoculated with wild-type rhizobia but not in plants interacting with rhizobia lacking nitrogenase activity. These results indicate that nitrogen fixation activates the expression of these iron deficiency-related peptides.

Furthermore, *ima1/2* knockout plants develop twice more nodules than wild-type plants. However, they are smaller and have reduced nitrogenase activity, indicating that IMA peptides might have a role in the nodule development program. Through grafting experiments, the authors concluded that IMA peptides might act systemically to regulate the number of nodules and to promote iron uptake for the nitrogen fixation process. Interestingly, the authors demonstrated that the expression of IMA peptide-encoding genes is regulated by the transcription factor NIN, which is another link to the symbiotic role of these peptides. The overexpression of IMA1 and IMA2 led the authors to confirm that these peptides participate in iron uptake and transport in nodules. Finally, the authors found that nitrogen sources increase the expression of IMA1 and IMA2, which suggests that these peptides might play a role in the iron-nitrogen balance. This hypothesis was further validated through different physiological analyses in knockout *ima1/2* *L. japonicus* and *A. thaliana* plants.

Indeed, the data presented in this study will contribute to understanding how root nodules obtain iron to sustain the nitrogen fixation process and provide evidence of how plants control the iron-nitrogen balance, ensuring an optimal development.

Comments:

- 1) The authors state that "... little is known about the molecular mechanism of how Fe signaling is activated and how Fe is provided to nodules during RNS". This is not entirely true. It is known that NCR247 confiscates haem groups to promote iron uptake in *M. truncatula* nodules (Sankari et al. 2022. *Nat Microbiol.* 7: 1453-1465). Hence, the authors must mention this in the introduction section. Also, the authors must discuss this and the possibility that IMA peptides are essential for iron uptake in determinate nodules.
- 2) Page 3, line 102: Please indicate the tissues used for the RNA-seq analysis.
- 3) Page 3, lines 103-106: Please indicate the candidate genes' selection criteria.
- 4) The authors observed that knocking down IMA1 and IMA2 led to a 50% increase in nodules. Indeed, the authors suggest that these peptides might play a role in the autoregulation of nodulation pathway. This is an interesting observation. The authors can perform a few experiments to confirm this hypothesis. These experiments are: a) assess the expression of TML and miR2111 in the *ima1/2* knockout plants, expecting that the expression of miR2111 is higher than TML; b) overexpress IMA1/2 in transgenic roots from the *har1* mutant, expecting that overexpression of IMA1/2 reduce the number of nodules, and 3) treat *har1* mutant plants with different Fe concentration, expecting that higher concentrations of Fe reduce the number of nodules.
- 5) Through grafting experiments, the authors conclude that IMA1 and IMA2 peptides have a systemic effect on the number of nodules. However, they observed normal nodule formation when Ljima1/2 and WT were used as rootstock. Can IMA1/2 regulate this phenotype locally too?
- 6) The authors observed that nitrogen increases the expression of IMA1 and IMA2. Interestingly, this effect is no longer observed when the plants grow with higher Fe concentrations. With this and other experiments in *L. japonicus* and *A. thaliana*, the authors conclude that IMA peptides play a role in controlling the nitrogen-iron balance. This is an exciting observation; however, my main concern is that the authors use toxic Fe concentrations that plants might not encounter in nature.

How artificial is this effect? What is the Fe concentration used in their nutrient solution? Can you see the reduction of the expression of IMA using optimal physiological Fe concentrations?

7) The authors suggest that nitrogen might participate in the Fe uptake via the action of IMA peptides. Indeed, they suggest that products of nitrogen assimilation might participate in this process. Nitrate reductase is a crucial enzyme in the nitrogen assimilation process. Like nitrogenase, nitrate reductase requires Fe for optimal function. Is it possible that the high activity of nitrate reductase, and hence the high demand for iron, promotes the Fe uptake? It is essential to discuss this in detail.

8) Generally, it is essential to know the Fe concentration that the authors use as optimal or deficient iron concentration. This will help to understand the effect of nitrogen on Fe uptake.

9) Figure 1a: The authors must try to plot the expression levels in a box plot and show statistics. The current form is hard to follow what is mentioned in the main text.

Reviewer #2 (Remarks to the Author):

Internal nitrogen status of plants is prerequisite for symbiotic nitrogen fixation, and other nutrients (like Fe) required for symbiotic nitrogen fixation also affects the rhizobia-legume symbiosis. However, the regulatory network underlying this complex biological process is largely unclear. In this study, Ito et al used a nice design to identify LjIMA1/3 as important regulators of root nodule symbiosis and nitrogen homeostasis in response to internal nitrogen status. They found that LjIMA1/3 expression was induced by symbiotic nitrogen fixation and external nitrogen and Fe deficiency. They generated single and double mutants of LjIMA1/3 and found these peptides-coding genes play a positive role in Fe uptake during the establishment of RNS. Furthermore, they found that LjIMA1 expression is regulated by the key transcription factor, LjNIN, uncovering a direct molecular link between the Fe acquisition mechanism and a key nodulation signaling pathway. Finally, they found that IMA peptides have conserved roles in regulating nitrogen homeostasis by adjusting nitrogen-Fe balance in both *L. japonicus* and *Arabidopsis thaliana*. These findings expanding our understanding of interaction between IMA-mediated Fe signaling and nitrogen-related physiological processes. The results are interesting, and the manuscript is well written. But there are still some questions need to be addressed.

Major comments:

1. The title should be revised to properly cover all the information, such as the function of IMAs in iron.
2. For the transcriptome assay, they aimed to find the genes that respond to internal nitrogen status, so they used shoots at 11 DAI inoculated with WT, Δ NifH rhizobia or non-inoculated shoots. They should emphasize whether there is any nitrogen in the growth media and provide evidence that the shoots in three different conditions indeed have different internal nitrogen contents in Fig. S1. Analysis of nitrogen fixation efficiency of these nodules should be included. In addition to nodule phenotype, the shoots of these plants (what these plants look like) should be shown.
3. Although the authors payed attention on the genes encoding small peptides, the transcriptome data is quite important. I suggest the authors to show the overall expression comparison of the genes and biological processes under three internal nitrogen conditions.
4. It was reported that AtIMAs control shoot-to-root signalling of Fe deficiency. LjIMA1/2 may have similar role in controlling iron uptake during RNS and that would be the reason that the reciprocal graft of Ljima1/2 to WT restored Ljima1/2 phenotype in nodulation. The authors need to measure Fe content and LjIRT1/LjFRO2 expression in the grafted plants.
5. LjNIN regulates LjIMA1 expression: for the binding of LjNIN to the LjIMA promoter, the authors only showed EMSA result. As EMSA usually shows the in vitro protein-DNA binding with high false positive rate, a ChIP experiment for LjNIN and LjNLP1 or LjNLP4 in vivo binding to LjIMA1 promoter is suggested. In the transactivation assay, more controls are needed, including the mutant LjIMA1 promoter and LjNLP1 or LjNLP4.
6. It is apparent that LjIMA1/2 have roles in both early infection and nodule formation stages. The phenotypic analysis of the Ljima mutants should be performed and the expression patterns of both genes at 3 dai (correlating to LjNIN function, line 150) should be analyzed.
7. LjIMA1/2 was found in transcriptome data because they showed varied expression in the defective nodules. The authors proposed that these genes are induced by internal nitrogen. I

wonder if the levels of genes were induced to the WT levels when the defective nodules treated with external nitrogen.

8. Fig 5, the author tested the expression of LjIMA1/2 in plants treated with 5 mM KNO₃, while in the Fig S6 they analyzed the phenotypes under 1 and 10 mM KNO₃ treatments, how about the expression levels of LjIMA1/2 under these conditions? Is the induction of LjIMA1/2 by nitrogen in a dosage manner?

9. Line 205-206, please provide more evidence for "Ljima1/2 mutation attenuates nitrate-dependent plant growth and nitrate assimilation." nitrate content in Ljima1/2 was increased under nitrate treatment, how about the ammonium content?

10. Line 228-230, I do not agree with this, in fact from the Fig. 6a, the result showed that the shoot FW of atima3 in 50 mM nitrate conditions was comparable to that of WT, while in 5 mM nitrate treatment, atima3 shoot FW was higher than that of WT. It is confusing because AtIMA3 was induced by 50 mM but not by 5 mM nitrate.

11. The authors conclude that IMA peptides generally regulate nitrogen homeostasis by adjusting the nitrogen-Fe balance. This is interesting and the data in plant growth response to varied concentrations of N and Fe are convincing. How about nodulation?

12. Cellular and tissue response of LjIMA using LjIMA promoter:GUS lines to different combinations of N and Fe would support their conclusion about the role of IMA in N-Fe balance. Is LjLMA induced by Δ NifH rhizobia?

Minor concerns:

1. Line 136-138. The expression of LjIMA1/2 in the root at 11 dai, as well as in the shoot, was likely dependent on nitrogen fixation because Δ NifH rhizobia did not contribute to the gene expression at this stage. This description is not correct, in the figure S4c, the expression of LjIMA1 in the root inoculated with Δ NifH is also increased compared with 0 DAI

2. Fig. S5e, the result showed that low Fe had no obvious effect on mature nodule number, how about the total nodule number?

Reviewer #3 (Remarks to the Author):

In the article "IRON MAN peptides regulate root nodule symbiosis and nitrogen homeostasis in response to internal nitrogen status", the authors mainly discovered two peptide genes, LjIMA1 and LjIMA2, play a systemic role in concentrating internal Fe to the nodule, which is required for the establishment of RNS, and IMA peptides have conserved roles in regulating nitrogen homeostasis by adjusting nitrogen-Fe balance. This paper presents a new clue on iron provision and synergistic effect of iron and nitrogen during RNS, which is significant for further understanding on the regulation of the iron and nitrogen homeostasis in nodulation. The paper is logical and well written, but there are still a few flaws need to be solved which are listed below:

1. The introduction part needs to be enriched. For example, several transporters and transcription factors involved in iron transport during RNS were found. These previously findings need to be summarized better. The former studies of IRON MAN genes in Arabidopsis and other plants need to be introduced as well if available.

2. L141 to L153, authors proved LjNIN regulates LjIMAN1 expression. The phenotype of LjIMAN overexpression line in Ljnin mutant background is better to be show here.

3. L266, authors make a hypothesis that the appearance of NBS on LjIMA1 promoter allows it to be regulated by LjNIN. Please show the evidence that the IMA promoters in other "non-noduled" plants such as Arabidopsis, have no NBS.

4. L149, "the Ljnin mutation suppressed..." is better to be described as "the mutation on LjNIN suppressed..." or "in Ljnin mutant...was suppressed".

5. Ljnin mutant in L149 and Fig.3 was also described as Ljnin-9 in L286, please make them consistent.

6. The gene ID need to be show along with gene name either on the first time it appears or in the supplementary data.

7. References need to be cited accurately. For example, Line286, "Ljnin-9 plants was published

previously 34". But in the paper of Ref 34, there is very limit information about this mutant and the description of Ljnin-9 was pointed to another paper.

8. Some grammar mistakes need to be corrected. For example,

Line 45, "IRON MAN (IMA) peptide genes are expressed by symbiotic nitrogen fixation", "by" need to be replaced by "during" or "in". Similar mistake in L108 "which were expressed in the shoot by symbiotic nitrogen fixation".

Line 194-195, "LjIMA1/2 were found to be expressed in the shoot by nitrogen nutrients" should be described as "the expression of LjIMA1/2 were induced by nitrogen nutrients in the shoot".

Response to Reviewer #1

Iron is an essential micronutrient to sustain numerous biochemical processes, including symbiotic nitrogen fixation. Hence, plants tightly regulate its transport, distribution, and homeostasis according to environmental iron availability. The transcription factor PYE is a crucial genetic component in regulating plant responses to iron deficiency. PYE activates the expression of numerous genes required to cope with this nutritional status. Once plants fulfill their iron needs, the E3 ubiquitin ligase BRUTUS, an iron sensor, interacts with PYE, promotes its degradation, and inhibits plant responses to iron deficiency.

IRON MAN (IMA) peptides are synthesized under iron-deficient conditions and interact with BRUTUS, which allows PYE to activate the expression of the iron-deficiency-responsive genes. Hence, IMA peptides play a crucial role in iron uptake. Despite this progress in our understanding of how plants cope with iron deficiency, we still need to understand how root nodules obtain the appropriate concentration of iron to sustain the symbiotic nitrogen fixation process. In this manuscript, the authors performed different experiments to answer this question experimentally in the model legume *Lotus japonicus*. The authors found that the expression of LjIMA1 and LjIMA2 is increased in shoots and roots from plants inoculated with wild-type rhizobia but not in plants interacting with rhizobia lacking nitrogenase activity. These results indicate that nitrogen fixation activates the expression of these iron deficiency-related peptides.

Furthermore, *ima1/2* knockout plants develop twice more nodules than wild-type plants. However, they are smaller and have reduced nitrogenase activity, indicating that IMA peptides might have a role in the nodule development program. Through grafting experiments, the authors concluded that IMA peptides might act systemically to regulate the number of nodules and to promote iron uptake for the nitrogen fixation process. Interestingly, the authors demonstrated that the expression of IMA peptide-encoding genes is regulated by the transcription factor NIN, which is another link to the symbiotic role of these peptides. The overexpression of IMA1 and IMA2 led the authors to confirm that these peptides participate in iron uptake and transport in nodules. Finally, the authors found that nitrogen sources increase the expression of IMA1 and IMA2, which suggests that these peptides might play a role in the iron-nitrogen balance. This hypothesis was further validated through different physiological analyses in knockout *ima1/2* *L. japonicus* and *A. thaliana* plants.

Indeed, the data presented in this study will contribute to understanding how root nodules obtain iron to sustain the nitrogen fixation process and provide evidence of how plants control the iron-nitrogen balance, ensuring an optional development.

Thank you for reviewing and evaluating our manuscript. According to your comments, we

performed additional experiments and revised the manuscript.

Comments:

1) The authors state that "... little is known about the molecular mechanism of how Fe signaling is activated and how Fe is provided to nodules during RNS". This is not entirely true. It is known that NCR247 confiscates haem groups to promote iron uptake in *M. truncatula* nodules (Sankari et al. 2022. Nat Microbiol. 7: 1453-1465). Hence, the authors must mention this in the introduction section. Also, the authors must discuss this and the possibility that IMA peptides are essential for iron uptake in determinate nodules.

In the revised version, we enriched the introduction with a description of Fe signaling mechanisms in general and Fe transport mechanisms in nodules (lines 79-112). In the discussion, we mentioned NCR247 and described the potential mechanism of Fe uptake via LjIMA in determinate nodules (lines 325-331).

2) Page 3, line 102: Please indicate the tissues used for the RNA-seq analysis.

We mentioned that 11 dai whole shoot tissues were used for RNA-seq analysis (lines 131-133).

3) Page 3, lines 103-106: Please indicate the candidate genes' selection criteria.

We added the candidate genes' selection criteria (lines 135-137).

4) The authors observed that knocking down IMA1 and IMA2 led to a 50% increase in nodules. Indeed, the authors suggest that these peptides might play a role in the autoregulation of nodulation pathway. This is an interesting observation. The authors can perform a few experiments to confirm this hypothesis. These experiments are: a) assess the expression of TML and miR2111 in the *ima1/2* knockout plants, expecting that the expression of miR2111 is higher than TML; b) overexpress IMA1/2 in transgenic roots from the *har1* mutant, expecting that overexpression of IMA1/2 reduce the number of nodules, and 3) treat *har1* mutant plants with different Fe concentration, expecting that higher concentrations of Fe reduce the number of nodules.

Thank you for your valuable comments about the potential relationship between IMA and AON and for suggesting interesting experiments. The increased nodule number in *Ljim1/2* suggests a possible association with AON. In *Ljima1/2* plants, the expression of MIR2111 was increased in the shoot and the expression of its target gene *TML* in the root was decreased (Fig. Xa,b). Overexpression of *LjIMA1/2* reduced the number of nodules in *har1* (Fig. Xc), although the effect was not as pronounced as we overexpressed *LjIMA1/2* in WT. Furthermore, a reduction in the number of mature nodules was observed by the supply of Fe in *har1* (Fig. Xc,d). These results support the possibility of the implication of LjIMA1/2 and Fe in AON. However, these data are preliminary and we think that they should not be included in this paper. We would like to investigate the detailed regulatory mechanisms between LjIMA1/2, Fe and AON, as a follow-up

paper to this study with more analyses.

Fig. X. Potential relationship between LjIMA1/2 and AON. **a.b.** RT-qPCR analysis of shoot miR2111 and root *TML* expression in WT and *Ljima1/2*. Shoots or roots of 5 dai plants were collected (n = 3 independent pools of shoot or roots derived from three plants). **c.** Total nodule number of *har1* plants with transgenic hairy roots carrying EV, *LjUBQ_{pro}:LjIMA1* or *LjUBQ_{pro}:LjIMA2* constructs 21 dai. **d.e.** Total and mature nodule number of *har1* plants treated with 1 mM Fe 14 dai. Fe (III)-EDTA was used for the Fe source. Scatterplots show individual biological replicates as dots. In **a.b.**, bars indicate mean values. Different letters indicate statistically significant differences ($P < 0.05$, one-way ANOVA followed by multiple comparisons). Asterisks indicate a statistically significant difference ($P < 0.05$, by a two-sided Welch's *t* test). ns means not significant.

5) Through grafting experiments, the authors conclude that IMA1 and IMA2 peptides have a systemic effect on the number of nodules. However, they observed normal nodule formation when Ljima1/2 and WT were used as rootstock. Can IMA1/2 regulate this phenotype locally too?

Yes, the result of the grafting experiment suggests that in addition to their systemic roles of LjIMA1/2 in the shoot, LjIMA1/2 in the root locally function to regulate RNS (lines 160-161).

6) The authors observed that nitrogen increases the expression of IMA1 and IMA2. Interestingly, this effect is no longer observed when the plants grow with higher Fe concentrations. With this and other experiments in *L. japonicus* and *A. thaliana*, the authors conclude that IMA peptides play a role in controlling the nitrogen-iron balance. This is an

exciting observation; however, my main concern is that the authors use toxic Fe concentrations that plants might not encounter in nature. How artificial is this effect? What is the Fe concentration used in their nutrient solution? Can you see the reduction of the expression of IMA using optimal physiological Fe concentrations?

Indeed, this study sometimes uses high Fe concentrations that may not exist in nature. We adopt the 1 mM Fe concentration for several experiments using *L. japonicus* because it has many research advantages; in our growth conditions, 1 mM Fe is not a toxic concentration to *L. japonicus*, as it has no negative effect on the growth of plants, and can fully rescue the *Ljima1/2* phenotypes (Fig. 4d-f and Supplementary Fig. 6b). Fe is supplied to *L. japonicus* plants in the form of regulated concentrations in B&D solutions, replacing the moisture in vermiculite with the solutions. The *LjIMA1/2* expression is reduced in the shoot by a physiologically plausible Fe concentration such as 0.1 mM (Supplementary Fig. 8).

7) The authors suggest that nitrogen might participate in the Fe uptake via the action of IMA peptides. Indeed, they suggest that products of nitrogen assimilation might participate in this process. Nitrate reductase is a crucial enzyme in the nitrogen assimilation process. Like nitrogenase, nitrate reductase requires Fe for optimal function. Is it possible that the high activity of nitrate reductase, and hence the high demand for iron, promotes the Fe uptake? It is essential to discuss this in detail.

Thank you for your important point. We mentioned your idea in the discussion (lines 365-369), which we would like to investigate the possibility in a future study independent of this paper.

8) Generally, it is essential to know the Fe concentration that the authors use as optimal or deficient iron concentration. This will help to understand the effect of nitrogen on Fe uptake.

This comment is related to 6). In the experiments using *L. japonicus*, which comprise most parts of this study, under our growth conditions, *L. japonicus* WT plants do not show any obvious symptoms of Fe deficiency in the absence of external Fe (0 mM Fe conditions), nor symptoms of Fe excess even at a concentration of 1 mM Fe. The apparent normality of the plant growth in the absence of Fe may be due to the use of internal Fe stored in the seed and/or Fe originally contained in vermiculite. Therefore, *L. japonicus* has a wide range of Fe concentrations that it can adapt to. Thus, it may be difficult to determine the optimal and deficient Fe concentrations. They may also vary depending on the developmental stages and physiological contexts. However, given that *LjIMA1/2* are expressed in response to plant Fe demand and their expression levels are Fe concentration-dependent, 0.1 mM, which reduces *LjIMA1/2* expression in the shoot, may be an optimal Fe concentration in the shoot during RNS (Supplementary Fig. 8). Meanwhile, *LjIMA1* expression is reduced by 0.5 mM Fe in the root (Supplementary Fig. 8), suggesting, different organs have different optimal Fe concentrations.

9) Figure 1a: The authors must try to plot the expression levels in a box plot and show

statistics. The current form is hard to follow what is mentioned in the main text.

Following your suggestion, Fig. 1a was modified as a box plot, in which the result of statistical analysis was included. In addition, we mentioned in the text that five *LjIMA* genes, namely, *LjIMA1*, *LjIMA2*, *LjIMA4*, *LjIMA5* and *LjIMA8*, are expressed during symbiotic nitrogen fixation (lines 137-141).

Response to Reviewer #2

Internal nitrogen status of plants is prerequisite for symbiotic nitrogen fixation, and other nutrients (like Fe) required for symbiotic nitrogen fixation also affects the rhizobia-legume symbiosis. However, the regulatory network underlying this complex biological process is largely unclear. In this study, Ito et al used a nice design to identify *LjIMA1/3* as important regulators of root nodule symbiosis and nitrogen homeostasis in response to internal nitrogen status. They found that *LjIMA1/3* expression was induced by symbiotic nitrogen fixation and external nitrogen and Fe deficiency. They generated single and double mutants of *LjIMA1/3* and found these peptides-coding genes play a positive role in Fe uptake during the establishment of RNS. Furthermore, they found that *LjIMA1* expression is regulated by the key transcription factor, *LjNIN*, uncovering a direct molecular link between the Fe acquisition mechanism and a key nodulation signaling pathway. Finally, they found that IMA peptides have conserved roles in regulating nitrogen homeostasis by adjusting nitrogen-Fe balance in both *L. japonicus* and *Arabidopsis thaliana*. These findings expanding our understanding of interaction between IMA-mediated Fe signaling and nitrogen-related physiological processes. The results are interesting, and the manuscript is well written. But there are still some questions need to be addressed.

Thank you for reviewing and evaluating our manuscript. According to your comments, we revised the manuscript with a substantial amount of new data.

Major comments:

1. The title should be revised to properly cover all the information, such as the function of IMAs in iron.

The title was revised.

2. For the transcriptome assay, they aimed to find the genes that respond to internal nitrogen status, so they used shoots at 11 DAI inoculated with WT, Δ NifH rhizobia or non-inoculated shoots. They should emphasize whether there is any nitrogen in the growth media and provide evidence that the shoots in three different conditions indeed have different internal nitrogen contents in Fig. S1. Analysis of nitrogen fixation efficiency of these nodules should be included. In addition to nodule phenotype, the shoots of these plants (what these plants

look like) should be shown.

We revised Supplementary Fig. 1 and added related sentences (lines 126-134). At 11 dai, the plants inoculated with Δ NifH rhizobia were defective in nitrogen-fixing activities (Supplementary Fig. 1b), but they showed no obvious differences in plant growth from non- or WT rhizobia-inoculated conditions (Supplementary Fig. 1a). In contrast, the causal relationship between defective symbiotic nitrogen fixation and plant growth was evident at 18 dai (Supplementary Fig. 1a).

3. Although the authors payed attention on the genes encoding small peptides, the transcriptome data is quite important. I suggest the authors to show the overall expression comparison of the genes and biological processes under three internal nitrogen conditions.

We showed the list of DEG and a summary of transcriptome analysis (Supplementary Dataset 1 and Supplementary Fig. 2).

4. It was reported that AtIMAs control shoot-to-root signalling of Fe deficiency. LjIMA1/2 may have similar role in controlling iron uptake during RNS and that would be the reason that the reciprocal graft of Ljima1/2 to WT restored Ljima1/2 phenotype in nodulation. The authors need to measure Fe content and LjIRT1/LjFRO2 expression in the grafted plants.

Grafting experiments show that shoot LjIMA1/2 systemically regulate RNS, while root LjIMA1/2 locally regulate RNS (Fig. 2a-c). In *L. japonicus*, grafting is labor intensive and it is not a realistic workload to prepare large quantities of samples to ensure a sufficient amount of dry weight to determine Fe content. Alternatively, we show shoot phenotypes and shoot chlorophyll contents of grafted plants (Fig. 2c,d). Since the amount of chlorophyll correlates with that of Fe, an increase in chlorophyll contents implies an adequate presence of Fe. Although *LjIRT1* and *LjFRO2* expression is induced by *LjIMA1/2* overexpression (Fig. 4a), as far as we examined, their expression is not induced during RNS and is independent of LjIMA1/2 (Supplementary Fig. 7). This suggests that a mechanism exists to collect Fe to the nodules without relying on LjIRT1 and LjFRO2.

5. LjNIN regulates LjIMA1 expression: for the binding of LjNIN to the LjIMA promoter, the authors only showed EMSA result. As EMSA usually shows the in vitro protein-DNA binding with high false positive rate, a ChIP experiment for LjNIN and LjNLP1 or LjNLP4 in vivo binding to LjIMA1 promoter is suggested. In the transactivation assay, more controls are needed, including the mutant LjIMA1 promoter and LjNLP1 or LjNLP4.

As you suggest, it would be desirable to perform ChIP and investigate whether NIN binds to the promoter of *LjIMA1* in vivo. However, as many researchers have experienced, ChIP of NIN remains difficult for unknown reasons and there have been few examples of successes. Indeed, we have tried in the past, but have not been successful enough. Given the time constraints, it may not be a good idea to challenge ChIP of NIN for many years until success is achieved. Here, we

would like to show the regulation of *LjIMA1* expression by LjNIN in other ways. The NRE to which NLP binds is structurally similar to NBS, and the EMSA results show that LjNLP1/4 binds to the NBS/NRE-like sequence on the *LjIMA1* promoter, albeit weakly than LjNIN (Fig. 3a). Following your suggestion, we used a mutated binding site of the NBS/NRE-like sequence in transactivation assay, which shows that reporter gene expression by LjNIN/LjNLP4 occurs in this cis-element dependent manner (Fig. 3b). We further show that overexpression of *LjNIN* results in the induction of *LjIMA1* expression (Fig. 3c,d). In addition, the level of rhizobia-inducible *LjIMA1* expression is lower in the *nin* mutant than in the WT (Fig. 3e). These results, while not perfect, suggest the possibility that *LjIMA1* is a direct target gene of LjNIN.

6. It is apparent that LjIMA1/2 have roles in both early infection and nodule formation stages. The phenotypic analysis of the Ljima mutants should be performed and the expression patterns of both genes at 3 dai (correlating to LjNIN function, line 150) should be analyzed.

We examined the number of infected threads at 5 dai and found that LjIMA1/2 may be involved in the regulation of rhizobial infection (Fig. 1f). The *LjIMA1/2* expression in *nin* mutant at 3 dai (Fig. 3e) suggests that LjNIN is required for *LjIMA1* and *LjIMA2* expressions. On the other hand, overexpression of *LjNIN* induced the expression of *LjIMA1*, not *LjIMA2* (Fig. 3c).

7. LjIMA1/2 was found in transcriptome data because they showed varied expression in the defective nodules. The authors proposed that these genes are induced by internal nitrogen. I wonder if the levels of genes were induced to the WT levels when the defective nodules treated with external nitrogen.

Following your comment, we performed the analysis and found that insufficient expression of *LjIMA1/2* due to deficient internal nitrogen was restored by external nitrogen (Supplementary Fig. 10).

8. Fig 5, the author tested the expression of LjIMA1/2 in plants treated with 5 mM KNO₃, while in the Fig S6 they analyzed the phenotypes under 1 and 10 mM KNO₃ treatments, how about the expression levels of LjIMA1/2 under these conditions? Is the induction of LjIMA1/2 by nitrogen in a dosage manner?

Following your comment, we performed the analysis and found that *LjIMA1* is expressed in a nitrate dosage-dependent manner (Supplementary Fig. 11e).

9. Line 205-206, please provide more evidence for “Ljima1/2 mutation attenuates nitrate-dependent plant growth and nitrate assimilation.” nitrate content in Ljima1/2 was increased under nitrate treatment, how about the ammonium content?

Further analysis showed neither nitrate nor ammonium treatment affected ammonium contents in the *Ljima1/2* plants (Supplementary Fig. 11d).

10. Line 228-230, I do not agree with this, in fact from the Fig. 6a, the result showed that the

shoot FW of *atima3* in 50 mM nitrate conditions was comparable to that of WT, while in 5 mM nitrate treatment, *atima3* shoot FW was higher than that of WT. It is confusing because *AtIMA3* was induced by 50 mM but not by 5 mM nitrate.

We agree with your comments and then repeated the experiments to reconsider the phenotypes of *Atima* plants. Plant growths of *Atima* at different nitrate concentrations were not as pronounced as those of *Ljima1/2* (Fig. 6a and Supplementary Fig. 11b), and we concluded that there were no remarkable differences among plants (Fig. 6a). On the other hand, *Atima1/2/3* tended to grow worse at 50 mM than at 5 mM. In addition, *Atima3* and *Atim1/2/3* showed a marked decrease in chlorophyll contents at 50 mM relative to 5 mM (Fig. 6b). These results are consistent with the fact that *AtIMA1/2/3* expression is induced by 50 mM nitrate (Supplementary Fig. 14a-f).

11. The authors conclude that IMA peptides generally regulate nitrogen homeostasis by adjusting the nitrogen-Fe balance. This is interesting and the data in plant growth response to varied concentrations of N and Fe are convincing. How about nodulation?

In Supplementary Fig 13, we examined nodulation phenotypes at varying concentrations of nitrogen and Fe. In the absence of Fe, consistent with previous reports, a high concentration of nitrate (10 mM) inhibited nodule formation in WT (Supplementary Fig. 13a-c). A low concentration of nitrate (1 mM) did not affect nodule formation in WT, but it attenuated nodule formation in *Ljima1/2* (Supplementary Fig. 13a-c), suggesting that *Ljima1/2* is hypersensitive to nitrate in RNS. In the presence of 1 mM nitrate, the nitrate levels in *Ljima1/2* accumulated to the same level as when WT was fed with 10 mM nitrate (Supplementary Fig. 13d,e). Thus, in *Ljima1/2*, excessive nitrate accumulation may be the cause of the inhibition of nodule formation. The hypersensitive response to nitrate and the excessive nitrate accumulation were restored by Fe supply (Supplementary Fig. 13). Therefore, it is likely that nitrate inhibition of nodulation is controlled by *LjIMA1/2*-mediated adjustment of nitrogen-Fe balance.

12. Cellular and tissue response of *LjIMA* using *LjIMA* promoter:GUS lines to different combinations of N and Fe would support their conclusion about the role of IMA in N-Fe balance. is *LjLMA* induced by Δ NifH rhizobia?

We provided the data of GUS activity of *LjIMA1_{pro}:GUS* plants in different N-Fe conditions, where Fe downregulated the *LjIMA1* promoter activity (Supplementary Fig. 12b). *LjIMA1/2* expression in RNS can be divided into two induction patterns: one in response to rhizobial infection and the other in response to nitrogen fixation. The *LjIMA1/2* expression in response to rhizobial infection at a relatively early stage, such as 5 dai, is induced by Δ NifH rhizobia as well as in WT rhizobia. At a relatively later stage of RNS, when nitrogen fixation occurs, such as 11 dai, the expression level of *LjIMA1/2* by Δ NifH rhizobia is much lower than WT rhizobia-inoculated condition (Supplementary Fig. 5a,b).

Minor concerns:

1. Line 136-138. The expression of LjIMA1/2 in the root at 11 dai, as well as in the shoot, was likely dependent on nitrogen fixation because Δ NifH rhizobia did not contribute to the gene expression at this stage. This description is not correct, in the figure S4c, the expression of LjIMA1 in the root inoculated with Δ NifH is also increased compared with 0 DAI

Due to the very high expression of *LjIMA1* at 11 dai roots, the previous version of the graph adopted log10 as the scale of the Y axis. We changed the Y axis to standard linear notation (Supplementary Fig. 5c). The statistical analysis using the current data set shows there is no difference in the expression of *LjIMA1* in the root between 0 dai and 11 dai Δ NifH (please see the source data).

2. Fig. S5e, the result showed that low Fe had no obvious effect on mature nodule number, how about the total nodule number?

We showed the data of total nodule number, in which different concentrations of external Fe did not affect the total nodule number (Supplementary Fig. 6e). Under our growth conditions, *L. japonicus* WT plants did not show any nodulation-related defects, such as low nodulation, resulting from Fe deficiency in the absence of external Fe (0 mM Fe conditions). The apparent normality of the plant growth and nodulation in the absence of external Fe may be due to the use of internal Fe stored in the seed and/or Fe originally contained in vermiculite.

Response to Reviewer #3

In the article “IRON MAN peptides regulate root nodule symbiosis and nitrogen homeostasis in response to internal nitrogen status”, the authors mainly discovered two peptide genes, LjIMA1 and LjIMA2, play a systemic role in concentrating internal Fe to the nodule, which is required for the establishment of RNS, and IMA peptides have conserved roles in regulating nitrogen homeostasis by adjusting nitrogen-Fe balance. This paper presents a new clue on iron provision and synergistic effect of iron and nitrogen during RNS, which is significant for further understanding on the regulation of the iron and nitrogen homeostasis in nodulation.

The paper is logical and well written, but there are still a few flaws need to be solved which are listed below:

Thank you for reviewing and evaluating our manuscript. According to your comments, we revised the manuscript.

1. The introduction part needs to be enriched. For example, several transporters and transcription factors involved in iron transport during RNS were found. These previously findings need to be summarized better. The former studies of IRON MAN genes in Arabidopsis and other plants need to be introduced as well if available.

In the revised version, we enriched the introduction with a description of Fe signaling mechanisms,

including the role of AtIMA peptides, and Fe transport mechanisms in nodules (lines 79-112).

2. L141 to L153, authors proved LjNIN regulates LjIMAN1 expression. The phenotype of LjIMAN overexpression line in Ljnin mutant background is better to be show here.

This comment is related to our claim that *LjIMA1* expression is regulated by LjNIN. We believe the cogency was enhanced by performing additional experiments (please see our response to major comments #5 from Reviewer #2). Overexpression of *LjIMA1* in WT causes a reduction in nodule number (Supplementary Fig. 6d). Since nodule formation is completely suppressed in the *Ljnin* mutant, we will not be able to discuss the effects of *LjIMA1* overexpression on nodulation.

3. L266, authors make a hypothesis that the appearance of NBS on LjIMA1 promoter allows it to be regulated by LjNIN. Please show the evidence that the IMA promoters in other “non-noduled” plants such as Arabidopsis, have no NBS.

Based on the results of additional experiments to show that, in addition to LjNIN, LjNLP4 binds to the NRE/NBS-like sequence on *LjIMA1* promoter, we modified this claim as "In the case of *LjIMA1*, the presence of an NLP-binding NRE on its promoter might have been a predisposition that facilitated the direct regulation by NIN (lines 353-355)". Since a recent paper has shown that NLPs regulate the expression of genes that act in Fe signaling (Ying et al., Mol Plant, 2023), it is entirely possible that LjNLP4 regulates *LjIMA1* expression.

4. L149, “the Ljnin mutation suppressed...” is better to be described as “the mutation on LjNIN suppressed...” or “in Ljnin mutant...was suppressed”.

Thank you for pointing this out. We corrected it.

5. Ljnin mutant in L149 and Fig.3 was also described as Ljnin-9 in L286, please make them consistent.

We made the description of *Ljnin* consistent.

6. The gene ID need to be show along with gene name either on the first time it appears or in the supplementary data.

We provided the gene ID in Supplementary Table 3.

7. References need to be cited accurately. For example, Line286, “Ljnin-9 plants was published previously 34”. But in the paper of Ref 34, there is very limit information about this mutant and the description of Ljnin-9 was pointed to another paper.

Thank you for pointing this out. We cited an appropriate paper (Suzaki et al., Development, 2012).

8. Some grammar mistakes need to be corrected. For example,

Line 45, “IRON MAN (IMA) peptide genes are expressed by symbiotic nitrogen fixation”, “by” need to be replaced by “during” or “in”. Similar mistake in L108 “which were expressed in the shoot by symbiotic nitrogen fixation”.

Line 194-195, “LjIMA1/2 were found to be expressed in the shoot by nitrogen nutrients” should be described as “the expression of LjIMA1/2 were induced by nitrogen nutrients in

the shoot”.

Thank you for pointing these out. We corrected them.

Reviewer #1 (Remarks to the Author):

Thanks very much for addressing all my comments. The revised version has significantly improved. Your preliminary data on the role of IMA peptides and iron on the Autoregulation of Nodulation (AON) pathways are encouraging. I have two suggestions: 1) The authors can keep the nodulation assays on har mutants, both overexpression of IMA genes and iron treatments. This data can back up their hypothesis about the role of IMA peptides on the AON, and the authors can send this data into the supporting information section and state that further investigation is needed to understand the mechanism of action of IMA peptides on this pathway. 2) The authors might remove the hypothesis about the role of IMA peptides in the AON pathway from the discussion. Although it will be an obvious question that the reader will have.

The authors have different evidence indicating that NIN controls the expression of IMA-encoding genes under non-symbiotic and symbiotic conditions. Is it possible that NIN participates in the iron uptake and iron-nitrogen balance? I am not asking for further experiments, but it is essential to at least discuss this possibility.

Page 5, Line 153: please indicate the percentage of increase.

Reviewer #2 (Remarks to the Author):

This is an interesting story and the revised manuscript have addressed all my concerns.

Reviewer #3 (Remarks to the Author):

The manuscript is well revised according to the reviewer's responses. I think it can be accepted for publication.

Response to Reviewer #1

Thanks very much for addressing all my comments. The revised version has significantly improved. Your preliminary data on the role of IMA peptides and iron on the Autoregulation of Nodulation (AON) pathways are encouraging. I have two suggestions: 1) The authors can keep the nodulation assays on har mutants, both overexpression of IMA genes and iron treatments. This data can back up their hypothesis about the role of IMA peptides on the AON, and the authors can send this data into the supporting information section and state that further investigation is needed to understand the mechanism of action of IMA peptides on this pathway. 2) The authors might remove the hypothesis about the role of IMA peptides in the AON pathway from the discussion. Although it will be an obvious question that the reader will have.

Thank you for reviewing and evaluating our manuscript. According to your comments, we added the data (Supplementary Fig. 9). The previous discussion was removed and a new discussion was added.

The authors have different evidence indicating that NIN controls the expression of IMA-encoding genes under non-symbiotic and symbiotic conditions. Is it possible that NIN participates in the iron uptake and iron-nitrogen balance? I am not asking for further experiments, but it is essential to at least discuss this possibility.

We discussed the points.

Page 5, Line 153: please indicate the percentage of increase.

We added the ratio of increase.

Response to Reviewer #2

This is an interesting story and the revised manuscript have addressed all my concerns.

Thank you for reviewing and evaluating our manuscript.

Response to Reviewer #3

The manuscript is well revised according to the reviewer's responses. I think it can be accepted for publication.

Thank you for reviewing and evaluating our manuscript.